# The Redox Revolution in Brain Medicine: Targeting Oxidative Stress with AI, Multi-Omics and Mitochondrial Therapies for the Precision Eradication of Neurodegeneration

**DOI:** 10.3390/ijms26157498

**Published:** 2025-08-03

**Authors:** Matei Șerban, Corneliu Toader, Răzvan-Adrian Covache-Busuioc

**Affiliations:** 1Puls Med Association, 051885 Bucharest, Romania; mateiserban@innbn.com (M.Ș.); razvancovache@innbn.com (R.-A.C.-B.); 2Department of Neurosurgery, “Carol Davila” University of Medicine and Pharmacy, 050474 Bucharest, Romania; 3Department of Vascular Neurosurgery, National Institute of Neurology and Neurovascular Diseases, 077160 Bucharest, Romania

**Keywords:** oxidative stress, neurodegenerative diseases, mitochondrial dysfunction, reactive oxygen species (ROS), antioxidant therapies, Nrf2 pathway, lipid peroxidation, protein carbonylation, multi-omics integration, precision redox medicine

## Abstract

Oxidative stress is a defining and pervasive driver of neurodegenerative diseases, including Alzheimer’s disease (AD), Parkinson’s disease (PD), and amyotrophic lateral sclerosis (ALS). As a molecular accelerant, reactive oxygen species (ROS) and reactive nitrogen species (RNS) compromise mitochondrial function, amplify lipid peroxidation, induce protein misfolding, and promote chronic neuroinflammation, creating a positive feedback loop of neuronal damage and cognitive decline. Despite its centrality in promoting disease progression, attempts to neutralize oxidative stress with monotherapeutic antioxidants have largely failed owing to the multifactorial redox imbalance affecting each patient and their corresponding variation. We are now at the threshold of precision redox medicine, driven by advances in syndromic multi-omics integration, Artificial Intelligence biomarker identification, and the precision of patient-specific therapeutic interventions. This paper will aim to reveal a mechanistically deep assessment of oxidative stress and its contribution to diseases of neurodegeneration, with an emphasis on oxidatively modified proteins (e.g., carbonylated tau, nitrated α-synuclein), lipid peroxidation biomarkers (F2-isoprostanes, 4-HNE), and DNA damage (8-OHdG) as significant biomarkers of disease progression. We will critically examine the majority of clinical trial studies investigating mitochondria-targeted antioxidants (e.g., MitoQ, SS-31), Nrf2 activators (e.g., dimethyl fumarate, sulforaphane), and epigenetic reprogramming schemes aiming to re-establish antioxidant defenses and repair redox damage at the molecular level of biology. Emerging solutions that involve nanoparticles (e.g., antioxidant delivery systems) and CRISPR (e.g., correction of mutations in SOD1 and GPx1) have the potential to transform therapeutic approaches to treatment for these diseases by cutting the time required to realize meaningful impacts and meaningful treatment. This paper will argue that with the connection between molecular biology and progress in clinical hyperbole, dynamic multi-targeted interventions will define the treatment of neurodegenerative diseases in the transition from disease amelioration to disease modification or perhaps reversal. With these innovations at our doorstep, the future offers remarkable possibilities in translating network-based biomarker discovery, AI-powered patient stratification, and adaptive combination therapies into individualized/long-lasting neuroprotection. The question is no longer if we will neutralize oxidative stress; it is how likely we will achieve success in the new frontier of neurodegenerative disease therapies.

## 1. Introduction

### 1.1. Defining Oxidative Stress and Its Role in Neurology

The brain, which is responsible for approximately 20% of oxygen consumption, is a delicate organ responsible for maintaining a balance between destruction and survival. This balance is dependent on a dynamic system governing the production of reactive oxygen species (ROS) and reactive nitrogen species (RNS) [1]. Under physiologic conditions, ROS and RNS are not only products of metabolism but also play roles in signal transduction, synaptic plasticity, and cellular defense [2]. When ROS or RNS are produced beyond the response of antioxidant defenses, oxidative stress is established and becomes a significant disruptor of cellular homeostasis. Oxidative stress disrupts molecular homeostasis in neurons, impairing their function and ultimately contributing to multiple neurodegenerative pathologies. ROS and RNS can be thought of as “double-edged swords”, where they are effective regulators and mediators of neuronal death or neuroregeneration [3].

There are aspects of oxidative stress inherent to neurons that make them extraordinarily susceptible to oxidative stress based on their unique metabolic and structural makeup. First, the brain has a significant metabolic oxygen requirement to operate effectively, resulting in the near-constant production of ROS through oxidative phosphorylation [4]. Second, neuronal membranes are enriched with polyunsaturated fatty acids (PUFAs) that are susceptible to lipid peroxidation [5,6]. Third, antioxidant defenses in the brain are limited at the cellular level, with a lower expression of enzymes like catalase compared to peripheral tissues [7]. Finally, neurons are postmitotic cells; they have limited cellular regeneration, and when damage accumulates over time, restoration becomes difficult and often leads to irreversible neurodegeneration. These characteristics, in conjunction with others, place the brain at the epicenter for pathologies associated with oxidative stress, and it is therefore critical to understand the underlying mechanisms in order to identify appropriate therapeutic strategies [8].

### 1.2. Overview of ROS, RNS, and Redox Homeostasis

Reactive oxygen species (ROS) are reactive oxygen-centered molecules which include both radicals (e.g., superoxide anion (O_2_^−^·) and hydroxyl radical (·OH)) and non-radicals (e.g., hydrogen peroxide (H_2_O_2_)). The superoxide is one major species of ROS that is generated through the leakage of electrons from the mitochondrial electron transport chain (ETC) (mostly at complexes I and III) during aerobic respiration. The superoxide is then quickly dismutated into H_2_O_2_ by the enzyme superoxide dismutase (SOD), which can then be detoxified by catalase or glutathione peroxidase (GPx). However, the H_2_O_2_ may, in the presence of the transition metal iron (Fe^2+^), undergo an extremely reactive Fenton reaction to produce the hydroxyl radical:**Fe^2+^ + H_2_O_2_ → Fe^3+^ + ·OH + OH^−^**

The hydroxyl radical is such a strong oxidant with a short half-life that it is one of the most destructive ROS in the cell, and it initiates chain reactions of lipid peroxidation which lead to the loss of membrane integrity and cellular function [9]. It is then the aldehyde byproducts of lipid peroxidation, cytotoxic compounds like malondialdehyde (MDA), and 4-hydroxynonenal (4-HNE) that are notorious for their ability to form covalent adducts with proteins and nucleic acids [10]. These adducts can lead to alterations in protein structure and enzymatic dysfunction, interfere with the replication and repair of nucleic acids, and ultimately lead to widespread oxidative damage and cellular dysfunction. Increased levels of these byproducts have been and continue to be directly associated with the pathogenesis of neurodegenerative disorders, signaling the important role of peroxidation in neuronal damage [11].

Mitochondria represent the principal source of cellular ROS under physiological conditions, primarily due to electron leakage during oxidative phosphorylation, along with other sources that add to the oxidative burden to cells, particularly during pathological processes. One major source of ROS is NADPH oxidases (NOX enzymes), specifically NOX2 in activated microglia, which is an important source of ROS related to neuroinflammation [12]. NOX2 catalyzes the generation of superoxide, generating oxidative stress and potentiating the chronic inflammatory state associated with Alzheimer’s and multiple sclerosis. Peroxisomes are another important source of ROS; they are a side effect of the β-oxidation of fatty acids, where local catalase normally acts on any excess H_2_O_2_ in peroxisomes. When there are no longer sufficient defenses against ROS formation within the peroxisome, the protective catalase function may fail [13]. Another important contributor to oxidative bursts is xanthine oxidase, which is also a significant part of ischemia–reperfusion injury, where re-oxygenation results in an oxidative burst of superoxide, and oxidative tissue damage to the area of the ischemic insult [14]. One last source of potential oxidative burden is the endoplasmic reticulum (ER), where ROS is produced from protein folding. Under more prolonged ER stress requiring more extensive disposal of misfolded proteins, excessive amounts of ROS may be generated, producing more disruptions to cellular homeostasis [15]. RNS are also, in many cases, equally as important in redox biology and oxidative injury as ROS. In particular, the nitric-oxide radical (NO·) produced by nitric oxide synthesizes (NOSs) is a key signaling molecule that regulates vascular tone, synaptic plastically and neurotransmitter dynamics, with elevated NO· levels due to oxidative stress being available to react with superoxide to form peroxynitrite (ONOO^−^), a highly reactive and damaging species of RNS [16]. Peroxynitrite has the potential to nitrate tyrosine residues of proteins, potentially affecting the activity of enzymes, and changing cellular signaling pathways. Nitration is linked to mitochondrial dysfunction and neuronal death in neurodegenerative diseases and RNS, the latter being significant contributors to cellular death in addition to ROS [17].

To counteract the damaging effects of reactive oxygen and nitrogen species (ROS and RNS) on tissue, cells have evolved a compartmentalized and multilayered antioxidant defense system. The first layer of defense is enzymatic, where superoxide dismutases (SODs) catalyze the dismutation of the highly reactive superoxide anion (O_2_^−^·) into significantly less reactive hydrogen peroxide (H_2_O_2_), and through this, terminate any oxidative chain reaction initiation which would ensue from the superoxide anion [8,18].

There are three isoforms of SOD in the human body, which include different metal cofactors but have distinct sites of subcellular localization, illustrating the spatially specialized function of SOD. SOD1, or Cu/Zn-SOD, is mostly found in the cytosol, though it is also present in the mitochondrial intermembrane space, where it entraps a small amount of superoxide that escapes from the electron transport chain into the intermembrane space. The intermembrane space is the ideal location of SOD1 as it closely resembles the environment around Complexes I and III where superoxide is produced as a byproduct of oxidative phosphorylation [19,20].

SOD2, or the Mn-SOD isoform, is strictly localized to the mitochondrial matrix, where it detoxifies the majority superoxide. The level of electron intensity and high amounts of endogenous metabolism and oxygen consumption produced by mitochondria could make the mitochondrial matrix an oxidative stress hotspot, and therefore, SOD2 could maintain mitochondrial redox integrity [21]. SOD3, which is the extracellular isoform, is secreted from a variety of cells into the interstitial and vascular extracellular spaces, where it can scavenge superoxide anions in the extracellular microenvironment. SOD3 is particularly important for protecting endothelial surfaces, the extracellular matrix, and vascular integrity against oxidative insult [22].

Once superoxide is dismutated, hydrogen peroxide (H_2_O_2_), if not inactivated, can participate in Fenton chemistry to produce highly reactive hydroxyl radicals (·OH). Hydrogen peroxide can be inactivated by the enzyme catalase (which is highly expressed in peroxisomes), but it can also be inactivated by glutathione peroxidases (GPx), which inactivate hydrogen peroxide (and lipid hydroperoxides) in the presence of reduced glutathione (GSH) as a co-substrate [23]. The catalytic axis of SOD, catalase, and GPx works to efficiently inactivate ROS in specific organelles to ensure redox homeostasis [24,25]. In addition to enzymes, cells also have an extensive array of non-enzymatic antioxidants, which act as an additional layer of defense. Some examples include glutathione, vitamin C (ascorbic acid), and vitamin E (α-tocopherol). GSH is the most abundant antioxidant in the cytoplasm of cells (intracellular); it is usually the starting point for signaling, and is primarily involved in compensating for oxidants in the cell. Glutathione can directly scavenge ROS, while also acting as a cofactor for GPx. Glutathione is critical because it dynamically reverses the reduced state (GSH) to an oxidized form (GSSG), and it is important for regenerating oxidized vitamin C and E while linking antioxidant capacity together and amplifying it [26]. Lastly, when genomic DNA sustains oxidative insults, cells elicit DNA repair systems such as base excision repair (BER). A key part of this is achieved by DNA glycosylases, which recognize the oxidatively damaged base (e.g., 8-oxoguanine) and excise it, thereby launching repair cascades to restore genome integrity and prevent mutations or apoptotic signaling. This protective layer is especially critical for non-dividing cells such as neurons, which may experience irreversible functional impairment or cell death as a consequence of DNA oxidative damage accumulation [27]. To facilitate understanding of the biochemical interplay between ROS production and detoxification, a schematic representation is provided in Figure 1.

### 1.3. Relevance of Oxidative Stress to Neurological Disorders

Oxidative stress is a common pathological occurrence seen among neurological disorders given that it causes damage to lipids, proteins, and DNA, leading to considerable cellular deregulation. For neurodegenerative disorders like Alzheimer’s disease (AD), Parkinson’s disease (PD), and amyotrophic lateral sclerosis (ALS), oxidative stress is both the cause and the progressor of the disease [28]. Specifically, in AD, lipid peroxidation and oxidative modifications to proteins both contribute to amyloid-β aggregation and tau hyperphosphorylation, which are hallmarks of the disease. Oxidative modification of α-synuclein in PD leads to an aggregation of Lewy bodies that disrupts dopaminergic signaling and causes neuronal death [29,30].

The predominant outcome of oxidative stress is lipid peroxidation, which produces toxic aldehydes like MDA and 4-HNE, resulting in protein adducts and a loss of membrane integrity. Mitochondrial dysfunction also results from oxidative stress via direct damage to mitochondrial DNA (mtDNA) and the inhibition of enzymes that drive oxidative phosphorylation, thus creating a loop of self-perpetuating ROS overproduction [31]. Similarly to mitochondrial toxicity, the blood–brain barrier (BBB) can also be compromise under oxidative conditions, wherein ROS degrade tight junction proteins, allowing for immune entry and promote neuroinflammation [32].

Oxidative stress mechanisms are also prominent in neurodevelopmental and psychiatric disorders. In autism spectrum disorders (ASDs), increased levels of biomarkers of oxidative stress are associated with maternal inflammation and prenatal exposure to environmental toxins. In psychiatric disorders like schizophrenia and bipolar disorder, oxidative damage leads to impaired modulation of neurotransmitters, mitochondrial dysfunction, and neuroplasticity alterations [33].

With the tremendous range of manifestations of neurodegenerative diseases and the burden they impart at a global level, the need for directed molecular understanding of neurodegeneration is evident. The Global Burden of Disease (GBD) for 2019 and 2021 showed that AD and other related dementias alone accounted for over 28.8 million disability-adjusted life years (DALYs), and that these comprise one of the top ten global causes of disease burden, impacting over 55 million people worldwide [34]. PD accounted for over 3.2 million DALYs, and infection-based contributions were seen, with a prevalence of over 80% since 2000. These numbers are poised to increase markedly as the global population ages, resulting in enormous stress on health care systems [35,36]. The roles of redox imbalance and subsequent oxidative damage as common mechanisms of action across various neurological disorders not only emphasize the involvement of oxidative damage in the pathophysiology of these disease states but also allow for a re-evaluation of oxidative stress as a crosscutting therapeutic and diagnostic avenue [37].

### 1.4. Objectives and Significance of This Review

The purpose of this review is to provide a timely elucidation of the molecular mechanisms through which oxidative stress contributes to the pathogenesis of major neurological disorders, with a focus on the integration of recent discoveries from original research published from 2024 onward. This will include highlighting disorder-specific redox mechanisms of action, biomarkers of oxidative stress damage, and advances in diagnostics, including the use of omics techniques and in vivo imaging methods.

This review intends to highlight the current gaps in the literature and suggest possible future clinical uses of treatment approaches targeting redox state balance in neurological disorders. For example, emerging treatment strategies such as gene-based treatments, new antioxidant platforms, and novel drug delivery systems established through the use of nanotechnology will be discussed in terms of their potential for translatability into clinical settings. Given that this review intends to act as a bridge for new clinically based data arising from preclinical research, pragmatic recommendations for innovative pathways to target oxidative stress and therapeutic uses in neurology will be presented.

## 2. Mechanisms of Oxidative Stress in Neurological Disorders

### 2.1. Biochemistry of Oxidative Stress

The onset of oxidative stress in the CNS represents a cascade of biochemical events that leads to cellular dysfunction and ultimately the potential for neurodegeneration. The production of ROS and RNS is associated with cellular functions, and excessive generation leads to lipid peroxidation, protein oxidation, and DNA damage. A major distinction needs to be made between basal physiological levels of ROS and RNS essential for cellular signaling and pathological levels which lead to chronic cellular damage, a chronic state of mitochondrial dysfunction, and ultimately cellular death [38].

#### 2.1.1. Reactive Oxygen Species and Reactive Nitrogen Species

The main source of excessive reactive oxygen species generation in neurons is the mitochondrial ETC, which is a major site for electron leakage in both diseased and healthy states. After electron leakage from complexes I and III of the ETC, superoxide anion (O_2_^−^·) is produced when the electron prematurely interacts with molecular oxygen. An important implication of mitochondrial dysfunction is an increased amount of electron leakage, which corresponds with increased production of superoxide [39]. Mitochondria are not the only contributors to oxidative damage. During neuroinflammatory responses, NOX enzymes, particularly NOX2 in activated microglia, become a potent source of superoxide production, creating an oxidative environment that perpetuates the oxidative damage. In neurodegenerative diseases such as PD, chronic activation of NOX2 causes oxidative injury to dopaminergic neurons and xanthine oxidase, which are upregulated during ischemia–reperfusion injury, contributing bursts of ROS that cause acute neuronal death following the injury [40].

Once superoxide is formed, it is dismutated by superoxide dismutase (SOD) into hydrogen peroxide (H_2_O_2_). Although H_2_O_2_ is less reactive than superoxide, in the presence of transition metals such as iron or copper, it can undergo Fenton chemistry to generate hydroxyl radicals (·OH), which are highly reactive species capable of inflicting severe oxidative damage to lipids, proteins, and nucleic acids, thereby impairing cellular function and viability. Hydroxyl radicals are particularly damaging because they initiate lipid peroxidation and they are highly reactive due to their extremely short half life. Hydroxyl radicals are indiscriminant in their targets and will react with most biological macromolecules [41]. Neurons are particularly sensitive to lipid peroxidation caused by oxidative stress because they use lipid-rich membranes that contain PUFAs. The breakdown of lipids by products formed from lipid peroxidation, including MDA and 4-HNE, disrupts cell membrane integrity and forms covalent adducts with proteins and DNA, which ultimately impair neuronal function [42]. Unlike the global perspective on oxidative insult in Section 1, we want to highlight that in the context of AD, 4-HNE-modified proteins adversely affect synaptic function and tau homeostasis, leading to worsening cognitive decline.

In addition to ROS, nervous injury is further exacerbated by RNS. Under normal physiological conditions, nitric oxide (NO·), a key signaling molecule produced by nitric oxide synthase (NOS), regulates vascular tone and synaptic transmission [43]. However, in models of neuroinflammation and during the pathogenesis of neurodegeneration, it is over-produced by inducible NOS (iNOS), a specific isoform of NOS, and interacts with superoxide to form the highly reactive oxidant species peroxynitrite (ONOO^−^). Peroxynitrite induces nitrative stress, which generates nitrotyrosine adducts on proteins. One major target of peroxynitrite is mitochondrial Complex I, and nitration of tyrosine can impair its activity and mitochondrial depolarization, which leads to lower ATP production [17]. In PD, this impairment can facilitate a cascade leading to the death of dopaminergic neurons. Additionally, nitration of α-synuclein facilitates aggregation into insoluble fibrils, which facilitates the formation of toxic Lewy bodies, the pathological hallmark of PD [44].

Alongside redox-driven processes, a number of studies have recently begun to indicate that the physicochemical cellular environment may promote the cascade of oxidative damage and protein misfolding, specifically in regard to specific divalent cations (e.g., calcium (Ca^2+^), copper (Cu^2+^), and zinc (Zn^2+^)), because the presence of such ions is known to significantly affect the structures and thus the relative propensity for the aggregation of amyloidogenic proteins (e.g., Aβ and α-synuclein) by favoring β-sheet-rich fibrillation [45]. It has been shown that, subsequent to redox imbalance, elevation of ionic strength in either extracellular or intracellular conditions may act synergistically to destabilize protein solubility and nucleate toxic aggregates with oxidative stress [46].

#### 2.1.2. Cellular Targets and Oxidative Damage

The damage potential of oxidative stress can be described as follows: by attacking multiple cellular components simultaneously, it can create wide-ranging damage. One of the most striking effects of ROS-mediated damage is lipid peroxidation, which is also one of the first and most prominent effects. In the case of neurons, hydroxyl radicals initiate the lipid peroxidation pathway by abstracting hydrogen atoms from the methylene groups of the PUFAs, ultimately forming lipid radicals [47]. These radicals are then propagated to create a chain reaction that generates reactive aldehydes including MDA and 4-HNE [48]. As described in Section 1, 4-HNE has the ability to form covalent protein adducts with synaptic proteins, which disrupts neurotransmitter release, leading to synaptic failure; this is especially relevant in AD.

Protein oxidation induces structural modifications such as carbonylation, nitration, and S-nitrosylation, which in turn impair protein function, stability, and cellular signaling. S-nitrosylation of parkin activity is one prominent example of the functions of S-nitrosylated protein, in which an E3 ubiquitin ligase is rendered inactive. Parkin activity is required to breakdown misfolded proteins, and therefore if S-nitrosylated in vivo, it will not degrade the protein; hence, S-nitrosylated parkin leads to misfolded α-synuclein accumulation and ultimately the presence of Lewy bodies in the case of PD. Oxidative protein modifications have also recently been demonstrated in ALS. In the case of ALS-oxidized SOD1, mutants misfold and aggregate, leading to motor neuron degeneration [49,50]. DNA damage occurs in the presence of an ROS through direct modification of the nucleic acids, leading to the production of single-strand breaks, double-strand breaks or oxidized base damage such as 8-hydroxy-2′-deoxyguanosine (8-OHdG). Importantly, oxidative damage to mtDNA is detrimental in neurons because the neurons have a high metabolic rate and limited cellular mechanism to deal with oxidative damage. In ALS, oxidative damage to mtDNA also compromises axonal transport, leading to muscle atrophy and progressive motor dysfunction [51].

#### 2.1.3. Antioxidant Defense Systems

To combat oxidative damage, neurons have developed a complex network of both enzymatic and non-enzymatic antioxidant systems. For example, SOD is essential for inactivating mitochondrial superoxide; furthermore, for the neuronal protection of bioenergetics by SOD2 (mitochondrial), SOD2 requires the action of catalase and GPx to reduce H_2_O_2_ and hence prevents hydroxyl radicals from being generated [52]. Recall that we briefly reviewed the antioxidant benefits of superoxide and hydrogen peroxide in Section 1. Here, we will focus on how GPx dysfunction has been linked with increases in oxidative stress in ALS, potentially representing one of the factors that makes motor neurons so susceptible to oxidative damage.

Along with enzymatic antioxidant systems, there are also many non-enzymatic antioxidants, such as GSH, vitamin C, and vitamin E. Depletion of GSH leads to the typical outcome of oxidative stress in PD; marked depletion leads to more mitochondrial dysfunction and to bigger amounts of ROS. There is evidence that Nrf2, the key transcriptional activator of antioxidant defenses, is inhibited in neurodegenerative diseases, but may also be a potential therapeutic target. With respect to new therapeutic strategies, some researchers have been exploring Nrf2 activators and/or therapies that (i.e., prodrugs) enhance GSH levels to restore redox balance to prevent oxidative damage [53].

### 2.2. Pathophysiological Mechanisms in the Nervous System

Therefore, while increased oxidative stress is a biochemical event in the nervous system, and should be remembered as just that, importantly, it begins a complex set of pathophysiological mechanisms that eventually result in neurodegeneration, neuroinflammation, and anatomical damage. Pathological processes that negatively impact bioenergetics are primarily initiated by increased oxidative stress, and they also include increased mitochondrial dysfunction, impaired cellular clearance/salvage systems (e.g., autophagy), BBB permeability, and inflammation [54]. The next subsection aims to describe the primary pathophysiological consequences of oxidative stress that dictate trajectories of the onset and progression of neurodegenerative diseases.

#### 2.2.1. Mitochondrial Dysfunction in Oxidative Stress

Mitochondrial dysfunction is a major component of the positive feedback loop of oxidative stress occurring in neurons. Mitochondria generate more than 90% of cellular ATP via oxidative phosphorylation, and they are the main source of ROS. Under pathologic conditions that promote mitochondrial dysfunction, ROS can be produced in amounts beyond normal levels by damaging the mitochondrial ETC. In particular, Complex I and Complex III are well established as the primary sites of electron leakage within the ETC, where premature electron transfer to molecular oxygen leads to the generation of superoxide anion (O_2_^−^·) [55]. Excess levels of ROS also produce mitochondrial regulators, such as PINK1 and Parkin, which are frequently mutated in pathologies, such as PD, preventing the sampling and removal of damaged mitochondria (mitophagy). This increased ROS effect, i.e., damage, is spread to neighboring mitochondria [56].

If there is one area in which ROS can create permanently damaging effects on mitochondria, it is in the mtDNA. To be brief, mtDNA has no histones that are protective against damaging agents, and it has significantly less of the same repair machinery that nuclear DNA has. ROS damage to mtDNA impedes a mitochondrion’s ability to produce the requisite proteins in the ETC; this means that there will be no ATP synthesis, and the function of neurons will be compromised. In AD, mtDNA damage has been shown to coincide with the reduced enzyme activity of Complex IV in the ETC, and also ultimately, with the arterial energy depletion of hippocampal neurons. In AD, one of the first biochemical alterations is an energy deficit, and this occurs prior to there being significant deposition of amyloid-β plaque [57]. Also, increased levels of ROS likewise open up the mitochondrial permeability transition pore (mPTP). The mPTP is a non-selective pore residing in the inner mitochondrial membrane. High permeability associated with the opening of the pore manifests in the loss of mitochondrial membrane potential (ΔΨm), as well as in the loss of cytochrome c, which promotes apoptosis through caspase-dependent apoptotic pathways. In an ischemic stroke, opening of the mPTP by ROS contributes to neuronal death from reperfusion injury, which in turn drives infarct growth. Attempts to pharmacologically intervene with mPTP inhibitors like cyclosporin A showed neuroprotective effects in preclinical stroke models [58].

A further significant effect of oxidative stress is the inhibition of peroxisome proliferator-activated receptor gamma coactivator 1-alpha (PGC-1α), a master regulator of mitochondrial biogenesis. In PD, reduced expression of PGC-1α inhibits mitochondrial renewal, promoting the accumulation of defective organelles and perpetuated oxidative stress. Activation of PGC-1α is currently being researched to restore mitochondrial homeostasis while reducing neurodegeneration [59,60].

#### 2.2.2. Impaired Autophagy and Proteostasis

Neurons grow and maintain proteostasis by performing effective autophagy to avoid the accumulation of damaged proteins and organelles. Unfortunately, oxidative stress inhibits the autophagic flux at different points, resulting in an inability to degrade toxic aggregates and, ultimately, advancing the disease state. ROS can oxidatively modify important autophagy proteins (e.g., Beclin-1, Atg5-Atg12 complex), which will inhibit autophagosome assembly [61]. In Parkinson’s disease, S-nitrosylation of the E3 ubiquitin ligase Parkin inhibits its ability to tag damaged mitochondria to be removed. This results in the accumulation of damaged mitochondria, perpetuating the accumulation of intracellular ROS [62]. The oxidative state can adversely influence one of the main signaling pathways responsible for the autophagic pathway, the mechanistic target of the rapamycin (mTOR) signaling pathway. Oxidative stress alters mTOR, resulting in unwanted activation, which in turn inhibits the initiation of autophagy [63]. Inhibition of autophagy results in the accumulation of the pathogenic α-synuclein protein in PD and amyloid-β in Alzheimer’s disease, as they form toxic aggregated forms that can functionally impair the neuron. Furthermore, impaired autophagic clearance can lead to the pathogenicity of hyperphosphorylated tau, creating neurofibrillary tangles in Alzheimer’s. This autophagic impairment, coupled with oxidative damage to lysosomes that release hydrolytic enzymes to degrade cytoplasm, can result in destruction of the neuron [64].

In ALS, oxidative damage and stress impair the clearance of altered SOD1 mutants, leading to toxic aggregates in motor neurons. Studies have shown that enhancing autophagic capability (e.g., mTOR inhibition such as rapamycin or pharmacological activators of chaperone-mediated autophagy) reduces aggregate levels and enhances neuron survival [65,66,67].

#### 2.2.3. Neuroinflammation and Oxidative Stress Interplay

Neuroinflammation and oxidative stress are associated phenomena that form a thick, harmful feedback loop that underlies neuronal injury, and once triggered, the ROS activate the redox-sensitive transcription factor nuclear factor-kappa B (NF-κB); the pro-inflammatory cytokines that NF-κB upregulates can include tumor necrosis factor-alpha (TNF-α), interleukin-1 beta (IL-1β), and interleukin-6 (IL-6) [68,69,70]. The pro-inflammatory cytokines activate and recruit microglia (the resident immune cells of the CNS), in leading to more ROS being produced from NADPH oxidase (NOX2) activation. In the case of multiple sclerosis (MS), the superoxide (from NOX2) generates oxidative damage to myelin basic protein (MBP), and it is oxidative damage that leads to demyelination and axonal injury [71].

With aging in AD, the acute microglial activation required to clean the amyloid-β plaque debris leads to chronic microglial activation with constant production of ROS and synaptic failure; when there are higher levels of 4-HNE and peroxynitrite (ONOO^−^) in microglia that are farther away from plaques, this suggests the presence of chronic neuroinflammation [72]. There is a consistent pattern in animal studies of neurodegeneration, although these studies are limited, that suggest that medications that inhibit microglial activation, such as NOX2 inhibitors and anti-inflammatory cytokines, reduce oxidative stress and improve cognition [73].

#### 2.2.4. Blood–Brain Barrier Integrity and Oxidative Stress

The BBB is a unique structure that protects the brain from the systemic circulation’s toxicants and that regulates the transport of nutrients and metabolites. Blood–brain barrier dysfunction may result from oxidative stress when endothelial cells and tight junction proteins that are involved in blood–brain barrier integrity are damaged. Some tight junction proteins including occludin, claudin-5, and ZO-1 are highly susceptible to post-translational modification via phosphorylation and enzymatic degradation induced by ROS, leading to the greater permeability of the BBB [74]. this Table 1 below intends to outline additional important biomarkers of oxidative stress, the diseases that they are neurologically relevant to, their biological meaning, and their detection methods, aimed at provindg a standardized reference with which to identify and recognize redox dysregulation clinically and in the research setting.

During ischemic stroke, endothelial cell oxidative stress increases BBB dysfunction by increasing cerebral edema and the infiltration of peripheral immune cells such as T-cells and monocytes into the brain parenchyma. BBB compromise leads to the ability of toxic proteins such as amyloid-β in AD to accumulate, with a loss of clearance mechanisms to allow this accumulation and further ROS production [92]. Endothelial nitric oxide synthase (eNOS) function, when under oxidative stress, impairs vasodilation, leading to inefficient cerebral blood flow, contributing to neuronal injury via ischemic-type neuronal cell death [93].

#### 2.2.5. Interaction with Other Cellular Processes

Oxidative stress integrates with a multitude of cell pathways to amplify neuronal injury and dysfunction. An example of this integration is with the intrinsic apoptotic pathway in which endothelial cell-derived ROS induce mitochondrial outer membrane permeabilization (MOMP) and, ultimately, cytochrome c release, which initiates caspase-3 activation and subsequent undesired neuronal apoptosis, in instances of PD and stroke [94]. ROS can induce excitotoxicity by enhancing the overactivation of N-methyl-D-aspartate (NMDA) receptors, increasing calcium influx, which ultimately causes mitochondrial dysfunction and initiates additional ROS production; excitotoxicity also accelerates ROS production in ALS via excitotoxicity-induced motor neuron degeneration [95].

Oxidative stress is also involved in impairing glucose metabolism, contributing to neuronal insulin resistance. When insulin signaling in the brain is defunct, the process of glucose uptake is compromised, resulting in an energy-deprived state that continues to enhance neurodegenerative processes. This impaired glucose metabolism is considered a component of type 3 diabetes (brain insulin resistance) that eventually influences AD-related pathology. Therapeutic options involve targeting glucose metabolism with insulin-sensitizing agents or targeting oxidative stress with antioxidant supplements in an effort to limit brain damage in any neurodegenerative disease [96].

## 3. Oxidative Stress and Major Neurological Disorders

Oxidative stress is the primary pathological feature of many neurological disorders and is one of the most significant contributors to neuronal damage through ROS/RNS overproduction, mitochondrial dysfunction, and diminished redox homeostasis. While oxidative stress may differ from condition to condition, oxidative stress still mediates lipid peroxidation, protein oxidation, and DNA damage, all of which lead to synaptic dysfunction, neuroinflammation, and neuronal death [97]. This section aims to provide an overview of the important molecular mechanisms and impact of oxidative stress in neurodegenerative diseases, cerebrovascular diseases, demyelinating diseases, neurodevelopmental disorders and psychiatric diseases.

### 3.1. Neurodegenerative Diseases

Neurodegenerative diseases are a leading cause of cognitive and physical decline across the world, and they involve the progressive loss of populations of neurons and a build-up of protein aggregates. While the neuropathology varies between disorders, a common feature is oxidative stress, which accelerates neurodegeneration by promoting mitochondrial dysfunction, protein misfolding, and chronic neuroinflammation. In cases of AD, PD, and ALS, oxidative damage to cellular macromolecules causes a loss of neuronal homeostasis or drives neurons to trigger resident apoptotic pathways. Understanding the contributory role of oxidative stress in each of these disorders can lead to the development of treatment outcomes to address redox imbalances in central nervous system tissues [98].

#### 3.1.1. AD

In AD, oxidative stress is not just a secondary consequence of the disease but rather an early pathological event leading to synaptic dysfunction, Aβ aggregation, tau hyperphosphorylation and ultimately cognitive decline. One of the most commonly cited mechanisms for this is the positive feedback loop created by Aβ aggregation and ROS production. The use of Aβ oligomers with mitochondrial translocase of outer membrane 20 (TOM20) leads to mitochondrial ETC activity changes, with increased electron leakage from complexes I and IV contributing to greater levels of superoxide anions (O_2_^−^·) which rapidly convert into hydrogen peroxide (H_2_O_2_) [99]. To exacerbate matters, as H_2_O_2_ accumulates, it reacts through various processes of Fenton chemistry which produce hydroxyl radicals (·OH) that can further damage lipids, proteins, and nucleic acids. In parallel, Aβ aggregates have been shown to activate NADPH oxidase 2 (NOX2), triggering rapid bursts of ROS in both microglia and neurons that exacerbate oxidative damage and neuroinflammation. In turn, these sequential bursts of ROS in microglia and neurons contribute to neuronal damage. The breadth of the literature confirms that Aβ-induced ROS production damages and influences mitochondrial health and leads to less ATP production in the hippocampal neurons necessary for memory coding and retrieval [100,101].

Oxidative stress continues to influence lipid peroxidation, which impacts phospholipids, preferentially DOHAs and other PUFAs, in neuronal membranes. Once the PUFAs undergo peroxidation, reactive aldehydes are produced, for example, MDA and 4-HNE. These by-products of peroxidation can mutually or covalently form an adduct with Aβ peptides, likely causing them to aggregate further and promoting the toxicity of the Aβ peptide. For example, Aβ combined with 4-HNE resulted in greater oligomer stability and neurotoxicity than ‘native’ Aβ [102]. Similarly to ROS, products of lipid peroxidation impair glutamate transporters (e.g., EAAT2), thereby producing excitotoxicity and damaging synapses. Of outmost importance is that oxidative damage is not limited to Aβ; tau can undergo the same ROS-induced oxidative, post-translational modifications, such as tyrosine nitration, and/or acetylation, leading to irreversible changes in tau and disrupting the proteolytic process of the proteasome, as it can no longer recognize and degrade tau. Further, the excitotoxicity pathways of ROS may lead to increased tau accumulation as kinases, such as glycogen synthase kinase-3β (GSK-3β), are activated to keep tau phosphorylated, leading to tau tangles. In summary, Aβ-induced ROS production, lipid peroxidation, and tau hyperphosphorylation represent a triplet of neurotoxic events and additively further the development of AD [103].

#### 3.1.2. PD

PD is primarily characterized by the degeneration of dopaminergic neurons in the substantia nigra pars compacta and the aggregation of α-synuclein in Lewy bodies. Oxidative stress lies at the heart of this neurodegeneration, as dopaminergic neurons are extremely vulnerable to oxidation. Indeed, dopamine metabolism is a source of ROS itself since dopamine can undergo auto-oxidation to form dopamine quinones and superoxide anions [104]. Under normal conditions, dopamine is sequestered in synaptic vesicles, where vesicular monoamine transporter 2 (VMAT2) prevents auto-oxidation. In PD, however, several factors promote impaired vesicular storage and dopamine metabolism dysregulation, resulting in excess cytosolic dopamine and an overabundance of ROS. The quinone derivatives created as a result of dopamine oxidation then create covalent adducts with proteins, most notably α-synuclein, that promote abnormal folding and aggregation, resulting in the formation of Lewy bodies [105]. The presence of these aggregates also impairs synaptic transmission and axonal transport, ultimately leading to neuronal dysfunction [106].

A second major source of ROS overproduction in PD is mitochondrial dysfunction; there is an extensive body of literature backing this, with major evidence for impaired mitochondrial Complex I activity. Mutations in mitochondrial regulators, such as PINK1 and Parkin, may impair mitophagy, leading to an increase in damaged mitochondria that may leak superoxide into the cytosol. Postmortem studies of PD patients have shown considerable oxidative damage to mtDNAl; they have also elucidated the mitochondrial proteins that impair ATP production and further amplify oxidative stress [107]. The highly reactive nitrogen species peroxynitrite (ONOO^−^)—resulting from the reaction of nitrous oxide with superoxide—mediates the nitrosylation of mitochondrial proteins and subunits of Complex I; in the context of inflammation, this nitrosylation limits the catalytic function of proteins [108]. The downstream ramifications of the above cascades eventually culminate in a subsystem destabilizing cytochrome c in the cytosol prior to activating apoptosis via caspase. Given the above, it is not challenging to envision the coupling effects of mitochondrial dysfunction → aggregation, including the promotion oxidative stress, with dopamine oxidation creating a feedback loop that sustains the degeneration of dopaminergic neurons in PD [109].

From a therapeutic perspective, assessing ROS/oxidative stress could have relevant impacts on PD. The antioxidant N-acetylcysteine (NAC) is a precursor of glutathione and is known to reduce combined ROS levels significantly and facilitate mitochondrial function, playing initiator roles, as identified in preclinical studies. Iron chelation by deferiprone is proposed to suppress the ROS production that results from Fenton chemistry via modulating iron build-up when there are excessive amounts of rail iron in dopaminergic neurons [110].

#### 3.1.3. ALS

ALS is characterized as the selective degeneration of motor neurons of the spinal cord, brainstem, and motor cortex; it can culminate in progressive muscle weakness, and ultimately respiratory failure. Oxidative stress has been considered in ALS, alongside the genetic familial forms of ALS and SOD1. In the familial forms of ALS, SOD1 mutations not only inhibit the enzymatic activity of SOD1 but provide a toxic gain of function through mutant SOD1 misfolding and through producing ROS rather than detoxifying the ROS [111]. When SOD1 is a misfolded mutant, the mutant SOD1 protein will likely bind directly to the mitochondrial membranes and interfere with the calcium homeostasis of mitochondria and promote mPTP opening. Mitochondria will release cytochrome c into the cytosol and initiate a distinguished cascade to apoptosis via caspases, according to [112]. Some of the destruction resulting from oxidative stress will act on neurofilament and axonal transport, and this is a major degenerative mechanism due to the long axonal projections of motor neurons. ROS will damage the axonal transport proteins dynein or kinesin and as a result will interfere with axonal transport between mitochondria and synapses, contributing to NMJ pathway dysfunction and muscle atrophy [113]. ROS will mediate lipid peroxidation in the plasma membrane of motor neurons and introduce aldehydes that are toxic to ion channels, thereby also converting any cytotoxicity into excitotoxicity. Neuroinflammation plays a critical role in ALS, with activated microglia and astrocytes releasing pro-inflammatory cytokines and generating ROS via NOX2 activation, thereby compounding the oxidative burden on motor neurons [114].

In addition to SOD1 mutations, oxidative stress is now recognized as a unifying pathophysiological mechanism that spans nearly all forms of ALS, including both familial and sporadic cases. Among non-SOD1 familial mutations, hexanucleotide repeat expansions in the C9orf72 gene constitute the most common genetic cause of ALS and frontotemporal dementia [115,116]. These expansions lead to the generation of toxic RNA foci and dipeptide repeat proteins (DPRs), particularly poly(GR), poly(PR), and poly(GA), through repeat-associated non-AUG (RAN) translation. DPRs such as poly(GR) and poly(PR) have been shown to accumulate within mitochondria and disrupt key mitochondrial processes, including Complex I-dependent respiration, ATP production, and mitochondrial calcium buffering [117]. These mitochondrial insults trigger increased ROS production, mitochondrial depolarization, and the activation of the mitochondrial unfolded protein response (mtUPR), ultimately initiating caspase-mediated apoptosis. Furthermore, poly(GR) has been reported to interfere with nucleolar function and ribosomal RNA synthesis, creating translational stress and amplifying the redox imbalance in motor neurons [118].

Similarly, mutations in TARDBP, encoding the DNA/RNA-binding protein TDP-43, underlie another substantial subset of ALS cases and are also tightly linked to redox dysregulation. In sporadic ALS and many familial forms of it, TDP-43 is mislocalized from the nucleus to the cytoplasm, where it aggregates and forms stress granules resistant to clearance. These cytoplasmic aggregates impair mitochondrial fission–fusion dynamics, inhibit axonal transport of mitochondria, and reduce the expression of mitochondrial antioxidant enzymes such as peroxiredoxins and SOD2 [119,120]. The result is a loss of mitochondrial integrity, increased leakage of electrons from the respiratory chain, and the augmented formation of superoxide and hydrogen peroxide. TDP-43 aggregates also bind to and sequester mRNAs responsible for antioxidant responses, including those encoding components of the Nrf2-ARE pathway, thereby suppressing endogenous defense mechanisms and worsening oxidative injury [121].

Mutations in FUS, another nuclear RNA-binding protein associated with familial ALS, produce similar downstream effects. ALS-linked FUS variants mislocalize to the cytoplasm, where they form aggregates that interfere with DNA damage recognition and repair pathways, notably base excision repair (BER) [122]. Impairment of DNA repair sensitizes motor neurons to ROS-induced genomic lesions, including 8-oxoguanine formation and strand breaks, which, when unrepaired, result in cell cycle re-entry, chromatin instability, and neuronal death. Moreover, mutant FUS disrupts mitochondrial trafficking and reduces mitophagy efficiency, leading to the accumulation of dysfunctional mitochondria as persistent ROS sources [97].

In both familial and sporadic ALS, oxidative stress is further amplified by the sustained activation of glial cells. Activated microglia and reactive astrocytes upregulate NADPH oxidase isoform NOX2, producing sustained bursts of superoxide and exacerbating neuroinflammation [123]. This non-cell-autonomous injury contributes to a chronic oxidative microenvironment that overwhelms the antioxidant buffering capacity of motor neurons. Additionally, pro-inflammatory cytokines such as TNF-α and IL-1β released by glial cells further stimulate mitochondrial ROS production and interfere with glutamate uptake by astrocytes, intensifying excitotoxicity [124].

Taken together, these converging mechanisms across ALS subtypes demonstrate that oxidative stress is not merely a byproduct of mitochondrial failure or protein aggregation, but a core driver of motor neuron degeneration. The central role of ROS and redox imbalance in ALS pathogenesis—regardless of initiating mutation—supports the rationale for broad-spectrum antioxidant therapies and redox-sensitive interventions as promising avenues for disease modification in both familial and sporadic ALS [125].

Therapeutic strategies to mediate oxidative stress in ALS include Edaravone, a free radical scavenger which reduces quinone toxicity in microglia, with borderline efficacy to prolong functional gains in ALS [126]. Parallel trials include the use of mitochondrial antioxidants like MitoQ, as well as mitochondrial-targeted antioxidants that will promote and/or restore mitochondrial health and thereby determine the viability and survivability of motor neurons as well [127].

### 3.2. Cerebrovascular Disorders

Cerebrovascular disorders, including ischemic stroke, are a leading cause of disability and mortality worldwide. Oxidative stress is a significant factor in the pathophysiology of cerebrovascular disease and increases tissue damage in both the ischemic and reperfused states [128]. Given the high oxygen consumption of the brain, its capacity for antioxidant defense, and the abundance of polyunsaturated fatty acids in neural membranes and cellular components, the brain is vulnerable to the detrimental effects of oxidative damage. Excessive quantities of ROS produced in the brain can injure neurons, glial cells, and endothelial cells, promoting lipid peroxidation, mitochondrial dysfunction, and the disruption of the BBB [129]. This section intends to describe the complex mechanisms by which oxidative stress promotes cerebrovascular injury and the emerging therapeutic strategies with which to modulate these pathways.

#### 3.2.1. Ischemic Stroke and Reperfusion Injury

Ischemic stroke occurs as a result of the occlusion of a cerebral artery, thereby depriving the brain tissue of oxygen and glucose, which causes bioenergetic failure (e.g., the production of insufficient ATP, due to the loss of glucose and oxygen) and oxidative damage. During ischemia, there is reduced oxygen availability, and thus there is a transition away from oxidative phosphorylation to produce ATP towards anaerobic glycolysis, which inevitably leads to lactic acid accumulation and acidosis—causing a depletion of ATP [130]. Bioenergetic failure leads to a collapse of mitochondrial membrane potential and the bioenergetic collapse of the ETC, which primarily occurs at complexes I and III where superoxide anion (O_2_^−^·) is produced. ROS is produced at low levels under normal physiological conditions; however, in ischemic tissue, the ischemic brain cannot sequester excess ROS production in the mitochondria and subsequently produces a rapid and uncontrolled burst of ROS, secondary to mitochondrial dysfunction, as well as other sources, such as the activation or upregulation of ROS-generating enzymes [131]. When reperfusion occurs, the brain receives a sudden influx of oxygen, generating a massive burst of superoxide, hydrogen peroxide (H_2_O_2_), and hydroxyl radicals (·OH). As described previously, NO is formed from L-Arg through a three-step enzymatic process called NO synthase (NOS) under the influence of the enzyme NADPH oxidase. The reaction of superoxide on L-Arg produces nitrogen dioxide (NO_2_). In a self-perpetuating way, NO_2_ reacts with lipid peroxyl-radicals (LOO•) and produces products like nitrosyl-lipids (NO-LP) and reactive nitrogen species (RNS) that can react with and initiate the further oxidation of lipids, proteins, and nucleic acids, leading to cellular injury and necrosis that is exacerbated during reperfusion [132,133].

Oxidative stress (ROS overload) from the aforementioned reaction leads to lipid peroxylation, forming highly destructive lipid peroxidation products that are self-perpetuating and damaging to neuronal membrane phospholipids. Lipid peroxidation eventually results in aldehydes 4-HNE and MDA, which are highly toxic, starting from the free radicals of oxidized polyunsaturated fatty acids (e.g., arachidonic acid, docosahexaenoic acid). These aldehydes can covalently attach to DNA and proteins, leading to the dysregulation of cellular function and eventually neuronal cell death [23,134]. Damage caused by ROS also leads to damage to calcium channels and pumps from oxidative stress. Damage from oxidative stress leads to neuronal excitotoxicity and an overload of calcium ions, a prerequisite for neuronal injury. Calcium overload also causes neurotoxic effects due to the stimulation of calpains, Ca^2+^-dependent proteases that degrade neuronal cytoskeletal proteins, or the stimulation of phospholipases that continue lipid peroxidation [135]. Calcium overload also opens up the mPTP, causing mitochondrial swelling, mitochondrial membrane failure, and the release of pro-apoptotic proteins such as cytochrome c. The release of cytochrome c activates caspase-3-dependent apoptosis, which is thought to be a major contributor to delayed neuronal cell death after stroke [136].

Finally, reactive species can also damage nuclear and mitochondrial DNA, promoting oxidative modifications to the base amino acids (e.g., base methylation and formation of 8-Hydroxy-2′-deoxyguanosine (8-OHdG)), and stimulate the activation of poly(ADP-ribose) polymerase-1 (PARP-1) through the inhibition of PARP [137]. Once PARP is activated, excessive activation of PARP will lead to hyper-cautious NAD^+^ and ATP depletion and eventually lead to cell death by a failure mechanism known as the PARP-mediated type of death, also called parthanatos. Parthanatos is important in the ischemic stroke paradigm because it occurs independent of caspase-3 activation and results in the necrotic death of tissues in the penumbral tissue of the ischemic core [138].

ROS will initiate pro-inflammatory cytokines such as tumor necrosis factor-alpha (TNF-α) and interleukin-1 beta (IL-1β) to be released, which will lead to the recruitment of immune cells to the area of injury [139]. The ROS will originate from infiltrating neutrophils and macrophages and maintain a vicious cycle of oxidative and inflammatory injury through a myeloperoxidase (MPO)-dependent reaction, for example, as well as the generation of peroxynitrite through a reaction involving superoxide and nitric oxide to produce nitrated tyrosine residues on proteins that can alter the function of proteins and therefore continue the cycle of oxidative injury [140]. New treatment approaches have been developed to interrupt the vicious cycle of ROS production or mitochondrial dysfunction. Mito-targeting antioxidants (e.g., MitoQ and SKQ1) accumulate in mitochondria, scavenging superoxide and repairing damage to the ETC caused by the oxidant [141]. Mitochondrial permeability transition pore (mPTP) inhibitors that manifest in the production of cyclosporin A are effective when they reduce infarct size and delay cell death. There is also increasing interest in Nrf2 activators that produce a number of antioxidant enzymes, including SOD, catalases, and glutathione peroxidase, that increase the innate antioxidant expression following reperfusion [142].

#### 3.2.2. Oxidative Stress in Blood–Brain Barrier Dysfunction

The BBB is a selective endothelial barrier for the protection of the brain to limit harmful substances and allow for the transfer of beneficial nutrients and metabolic waste products. It is well established that oxidative stress can compromise the BBB, which is a major contributor to secondary brain injury following ischemic stroke. The BBB requires that tight junction proteins are assembled properly (e.g., z0cludin, claudin-5, and zonula occludens-1 (ZO-1)) to maintain structural integrity, as these proteins control paracellular permeability [143]. Reactive oxygen species (ROS) disrupt these tight junctions through oxidative modification but also potentially by activating proteolytic enzymes, such as matrix metalloproteinases (MMPs) [144]. The phosphorylation and degradation of tight junction proteins ultimately leads to increased solute paracellular leakage, and with this, the entry of plasma proteins and immune cells into the brain. For instance, fibrinogen, a plasma protein that leaks into the brain under BBB injury, can activate microglia and promote neuroinflammation and oxidative stress. Infiltrating neutrophils release additional ROS including reactive nitrogen species (RNSs) such as peroxynitrite which nitrify tight junction proteins (to further destabilize the BBB) [145,146]. Endothelial cells are incredibly sensitive to oxidative insults, as they use the mitochondrial system to generate ATP. Subsequently, ROS can lead to the excessive inhibition of endothelial nitric oxide synthase (eNOS) to produce less nitrate, contributing to pre-vasoconstriction, hypoperfusion, and endothelial injury to neurons. ROS can also augment endothelial signaling by the RhoA/rock signaling pathway, leading to a nerve-damaging environment. Endothelial cell contraction, tight junction instability, and lesions result in vascular dysfunction with secondary cerebral edema injury-activated phenotypes of neuronal channel proteins [147,148].

To preserve BBB integrity, MMP inhibitors (e.g., doxycycline, and marimistatin) should be included to prevent the degradation of tight junction proteins and mitochondria-targeted antioxidants, as should mitochondria-protective agents and brain-protective treatment addressing ROS-mediated endothelial cell injury to diminish posthypoxic-reperfusion injury on mitochondria and apoptosis [149]. The possible therapeutic approaches include the use of Nrf2 activators, such as sulforaphane and bardoxolone methyl, which is being evaluated because of their effects on the antioxidant defenses of the endothelium and their reparative effects on tight junction proteins [150].

### 3.3. Demyelinating Disorders

Demyelinating disorders reflect disturbances to the myelin sheath, the lipid structure that surrounds and insulates the axon and improves the efficiency of nerve impulse transmission. The most commonly studied demyelinating disease is MS, a disease in which chronic inflammation and oxidative stress work together to induce myelin damage, axonal loss, and neurodegeneration. ROS and RNS play an integral role in stimulating and perpetuating the demyelination process by directly attacking oligodendrocytes, inducing mitochondrial dysfunction, and increasing neuroinflammation [151,152]. This section will discuss oxidative stress-related demyelination and potential treatment options for correcting redox imbalances.

In MS, oxidative stress contributes to myelin loss and oligodendrocyte death. Oligodendrocytes are myelin-producing and repair cells. The inflammatory state of the lesions in MS induces a rapid burst of ROS, primarily through NOX2 (NADPH oxidase) in the activated microglia and macrophages. The NOX2 generates superoxide anions, the NOX hyperoxidases it to yield hydrogen peroxide (H_2_O_2_), which diffuses throughout cell membranes and induces lipid peroxidation, and the subsequent by-products of lipid peroxidation such as MDA and 4-HNE covalently bind to proteins and lipids present in myelin, which gain steric hindrance, leading to a destabilized structure and disrupted signal-conductance [153,154]. Although they are not especially classified, oligodendrocytes are uniquely sensitive to oxidative stress compared to other cell types because of their metabolic activity and reliance on mitochondrial ATP production for myelin biosynthesis. In fact, ROS can directly assault the mitochondrial DNA (mtDNA), as transcription factors (TF) mitigate mitochondrial respiration chains that function to produce lipid precursors for myelin bioproduction. Excessive ROS attack results in mitochondrial failure through dysfunctionality and starvation damage [155]. The accumulation of ROS in the mitochondria stimulates the formation of superoxide, causes the opening of the permeability transition pore (mPTP), stimulates cytochrome c release from the mitochondria, and induces oligodendrocyte apoptosis via caspase-dependent pathways. Postmortem analyses of MS lesions have also identified increases in oxidatively damaged mtDNA in oligodendrocytes, coinciding with diminished ATP production and a loss of myelin [156].

Oxidative damage also extends to axons, and ROS-induced loss of myelin further exposes the underlying axonal membrane to injury [157]. The underlying membrane is now fully exposed to the ROS milieu, and more importantly, RNS. The RNS in question is peroxynitrite (ONOO^−^), which acts primarily through nitrating the tyrosyl residues of axonal proteins, leading to diminished axonal transport and conduction velocity. Peroxynitrite and the nitration of neurofilament proteins are further recognized to play causal roles in axonal degeneration and future disability in MS [158]. Another significant concept regarding oxidative stress in MS pertains to the disruption of iron homeostasis in the CNS. Excess iron accumulation generates a pro-oxidant environment in MS lesions, due to iron-mediated Fenton crude reactions, leading to the oxidation of H_2_O_2_ into highly reactive hydroxyl radical (•OH) intermediates [159]. The oxidative burst then extends to an attack on lipids and proteins, and, subsequently, DNA damage. It has also been reported that iron deposits found in microglia and oligodendrocytes are a major source of oxidative stress in chronic MS lesions, suggesting that dysregulated iron metabolism is a central variable in neurodegeneration in MS [160].

Oxidative stress also establishes an avenue to perpetuate neuroinflammation via activating redox-sensitive transcription factors like nuclear factor-kappa B (NF-κB). Once activated, NF-κB is responsible for inducing the expression of pro-inflammatory cytokines such as tumor necrosis factor-alpha (TNF-α) and interleukin-1 beta (IL-1β)—recruiting immune cells to the lesion and promoting heightened inflammation [161]. Activated microglia and astrocytes augment ROS production via NOX2 activation, potentially creating a feedback loop of increased oxidative and inflammatory damage. Over time, with the accumulation of neuroinflammation from cellular damage, this rate-limiting step ultimately leads to the secondary loss of axons, cortical demyelination, and disability as MS progresses [162].

Novel therapeutics have been introduced in the medical community to inhibit oxidative stress, with the hope of breaking the cycle of injury that keeps occurring in MS and that ultimately leads to debilitation. Antioxidants such as N-acetylcysteine (NAC), a precursor to glutathione, appear to reduce levels of ROS and, seemingly, oxidative damage in oligodendrocytes [163]. Also, there have been mitochondria-targeted antioxidants, such as MitoQ and SKQ1, that have been able to prevent oligodendrocyte apoptosis and protect mitochondrial function in experimental models of neurodegeneration. Antioxidants that target iron chelation will reduce the availability of free iron, and therefore reduce excess ROS levels from Fenton-mediated processes [164]. There are also approaches towards promoting the activation of Nrf2 to upregulate endogenous antioxidant defense and remyelination. Nrf2 promotes the expression of the detoxifying enzymes for glutathione, SOD, and catalase—all of which may have a role in protection, through reducing oxidative stress and promoting repair [165]. Another an exciting avenue of research includes dietary considerations and lifestyle changes aimed at reducing oxidative stress in MS patients. Diets high in antioxidants, for example, the Mediterranean diet, have correlated with reduced markers of oxidative damage and overall positive clinical outcomes [166]. Clinical trials continue to investigate polyphenols that are simultaneously antioxidant and anti-inflammatory, or those that slow disease progression, such as resveratrol and curcumin [167].

### 3.4. Neurodevelopmental Disorders

Neurodevelopmental disorders refer to a heterogeneous group of conditions which are characterized by abnormal brain development and associated dysfunctions in cognition, motor abilities, and social function. Oxidative stress is becoming a pertinent factor in the etiology of neurodevelopmental disorders, especially in ASD, where changes in ROS levels and antioxidant defenses are consistently identified with neurodevelopmental dysfunction [168]. The developing brain is particularly susceptible to oxidative stress due to its high metabolic rate and is made up of large amounts of polyunsaturated fats and immature antioxidant systems, making it susceptible to damage by ROS [169,170]. This subsection will discuss the role oxidative stress has in neurodevelopmental disorders, and more specifically, ASD, in terms of its mitochondrial dysfunction, environmental risk contributors and possible new therapeutic strategies.

Oxidative stress is a hallmark of ASD. Various studies have shown significantly increased biomarkers of lipid peroxidation, protein oxidation, and DNA oxidation in ASD. Studies of ASD identifying levels of lipid peroxidation biomarkers such as MDA and 4-HNE showed significantly elevated levels of these lipid peroxidation markers in the blood, urine, and brain tissue. Additionally, one oxidative DNA damage biomarker, 8-hydroxy-2′-deoxyguanosine (8-OHdG), was significantly increased in participants with ASD and correlated with an overall increasing severity of behavioral and cognitive symptoms. A simultaneous decrease in significant antioxidant defenses such as GSH, SOD and catalase would suggest impaired detoxification from ROS exposure, thereby increasing the overall oxidative burden [171]. Mitochondrial dysfunction is also a significant contributor to the development of the pathology of ASD and is believed to influence synaptic plasticity, the release of neurotransmitters, and neuronal survival. Abnormalities regarding mitochondrial structure and function have been found in many individuals with ASD, including reduced activities in the ETC complexes I, III, and IV, leading to inefficient ATP production and increased superoxide production [172]. Postmortem brain analysis has shown increased levels of oxidatively damaged mtDNA in the prefrontal cortex area and cerebellum, both critical areas in terms of cognitive development and motor skills. Because of ROS-induced mtDNA mutations, mitochondrial function is further compromised so a vicious cycle is created that includes the overproduction of ROS, energy deficiency, and ultimately neuronal dysfunction [173]. Additionally, oxidative stress impacts cytosolic calcium homeostasis, further de-polymerizing the mitochondrial membrane and resulting in apoptotic cascades during the development of neurons [174].

Environmental factors are prominent sources of oxidative stress in ASD, especially during prenatal development when the brain is most vulnerable to external injury. Prenatal stressors such as air pollution, pesticides, heavy metals (e.g., lead, mercury, arsenic) and endocrine-disruptive chemicals have been implicated in alterations of fetal antioxidant systems and increasing ROS generation [175]. For example, maternal exposure to airborne particulate matter has been linked to elevated ROS-induced oxidative damage in the fetal brain, possibly impacting the morphological development processes of neuronal differentiation and migration. Maternal metabolic conditions, such as gestational diabetes and obesity, also cause intra-uterine oxidative stress, increasing the amount of pro-inflammatory cytokines and ROS production [176,177]. Maternal hyperglycemia has also been found to result in ROS-mediated epigenetic changes to fetal neurons, resulting in the dysregulation of neural development genes involved in synaptogenesis [178]. Neuroinflammation is yet another contributor to oxidative stress in ASD, establishing a cycle of oxidative and inflammatory injury. Activated microglia produce reactive oxygen species (ROS) and inflammatory cytokines like tumor necrosis factor-alpha (TNF-α) and interleukin-6 (IL-6), which further enhance inflammation and oxidative damage to neighboring neurons and glia [179]. More importantly, elevated levels of oxidized proteins (nitrated and carbonylated proteins) impair synaptic functions through disruptions in receptor activity, vesicle trafficking and neurotransmitter release. More specifically, oxidative modifications to proteins appear to affect γ-aminobutyric acid (GABA)ergic signaling, which is well known to be impaired in ASD and amplifies hyperexcitability within neuronal circuits [180].

We have focused on the most recent, emerging therapeutics targeting oxidative stress in ASD, in both animal and human studies, that are aimed at restoring redox homeostasis and reducing ROS-induced toxicities. N-acetylcysteine (NAC), a precursor to glutathione, showed promising results in clinical trails in terms of improving irritability, repetitive behaviors, and social interaction among children with ASD [181]. NAC works by restoring intracellular GSH levels to increase the concentration of ROS scavengers and decrease oxidative damage. Similarly, coenzyme Q10 (CoQ10) is a vital component of the mitochondrial ETC and has antioxidant an function; studies on whether it can also improve mitochondrial function and reduce oxidative stress in children with ASD are ongoing. Additionally, certain dietary approaches, such as the Mediterranean diet, which is specifically rich in antioxidants and polyphenols, were associated with reduced biomarkers of oxidative damage and positive behavioral phenotypes [182,183,184]. There has also been growing interest in the use of natural compounds with antioxidant and anti-inflammatory properties to modulate oxidative stress and neuroinflammation in children with ASD. For example, sulforaphane and phenol compounds (which can be derived from cruciferous vegetables) have been effective in animal studies, activating the Nrf2 pathway, a transcription factor that enhances the expression of other endogenous antioxidant enzymes like superoxide dismutase, catalase, and glutathione peroxidase [185]. In preliminary clinical trials, individuals supplementing their diet with sulforaphane experienced improvements in social responsiveness, verbal communication and cognitive flexibility, thereby supporting their further development as a therapy [186]. Much of the current research into epigenetic reprogramming in ASD is focused on reversing gene modifications induced by oxidative stress. It is known that ROS induce DNA methylation, histone-modifying compounds, and non-coding RNA expression, which modify the transcription of genes critical for synaptic development and plasticity [187]. Experimental models involving DNA methyltransferase inhibitors and histone deacetylase inhibitors have indicated some potential for the restoration of the gene expression of receptor activity and improved behavioral phenotype. For example, studies have demonstrated that the BDNF gene is inhibited by ROS-induced hypermethylation, but when it is inhibited, the expression of BDNF is restored, and it promotes synaptic plasticity and neuroprotection [188,189].

Another relatively novel therapeutic approach considers the concept of the gut–brain axis. Most conditions associated with the gut–brain axis emerge from individuals with dysbiotic gut microbiota that perpetuate systemic inflammation and oxidative stress in the brain. The role of dysbiotic bacteria is that they produce lipopolysaccharides (LPSs) and other pro-inflammatory metabolites, producing immune responses and facilitating ROS entrance into the brain [190]. Clinical trials are underway to evaluate treatment with probiotics, prebiotics, and fecal microbiota transplantation (FMT) to reverse the dysbiotic state and improve gut homeostasis while reducing oxidative stresses in the brain. Promising early outcomes of probiotic and non-probiotic treatments have indicated alleviations of behavioral symptoms, anxiety, repetitive behavior and gastrointestinal issues in children with ASD [191].

### 3.5. Psychiatric Disorders

Treatments for psychiatric disorders, such as depression, schizophrenia, and bipolar disorder, are emerging as possibilities, being understood on the basis of redox dysregulation and oxidative stress. It is now well established that excessive ROS, lipid peroxidation products, and oxidized proteins are clearly associated with the presence of impaired neuroplasticity, mitochondrial dysfunction, and neurotransmitter dysregulation in psychiatric patients. The central nervous system’s susceptibility to oxidative stress, combined with chronic stress, inflammation, and environmental factors, renders oxidative stress the first and foremost contributor to psychiatric pathophysiology [192]. The following section describes the most commonly referenced mechanisms and treatment for oxidative stress in neuropsychiatric disorders.

#### 3.5.1. Depression

Major depressive disorder (MDD) is characterized by low mood, anhedonia, cognitive impairment, and altered neurovegetative functions that persist over time. There is mounting evidence to suggest that oxidative stress is a key factor that causes neuroplasticity impairment, mitochondrial dysfunction, and inflammation-promoting MDD. Studies typically identify altered levels of MDA, 4-HNE, and 8-OHdG plasma, cerebrospinal fluid, and postmortem brain tissues from MDD patients. Therefore, these biomarkers comprise evidence that MDD is correlated with significant damage to lipids, proteins and DNA that results in the degeneration of neuronal circuits located in the prefrontal cortex, hippocampus, and amygdala [193,194].

One of the most commonly referenced mechanisms that indicate the association of oxidative stress has been focused on BDNF signaling disruption. BDNF is critical for synaptic plasticity, long-term potentiation (LTP), and neurogenesis in the hippocampus [195]. Oxidative stress acts directly on BDNF expression in a manner dependent on altered ROS-induced DNA methylation and histone acetylation, which disrupt neuronal connectivity and survival. Lower levels of BDNF are associated with atrophy of the hippocampus, a structural correlate of MDD observed upon neuroimaging. In addition, BDNF levels also impair neurogenesis within the dentate gyrus, which contributes to cognitive impairments and emotional dysregulations [196]. Ongoing mitochondrial dysfunction is another key contributor to depression-related oxidative stress. Mitochondrial ETC Complex I and IV activity is decreased in the prefrontal cortex and hippocampus of MDD patients, leading to depleting ATP stores and increased ROS generation [197]. Mitochondrial dysfunction leads to ROS-mediated apoptosis by mitochondrial outer membrane permeabilization (MOMP) releasing cytochrome-c, which activates caspase-3-mediated cell death. Mitochondrial dysfunctional limitations are compounded by chronic stress and the related glucocorticoid-induced mitochondrial disruption. Increased glucocorticoids (GCs) lead to higher rates of ROS production and downregulated antioxidant systems like GPx, leading to continuous ongoing oxidative damage [198]. Oxidative stress also dysregulates the hypothalamic–pituitary–adrenal (HPA) axis, one of the diagnostic features of depression. Increases in ROS can lead to the impairment of glucocorticoid receptor (GR) signaling to diminish the feedback inhibition of the HPA axis from GR, causing the unabated production of cortisol. Continued HPA axis activation has increased oxidative stress and led to a self-perpetuating cascade, further exacerbating neuroendocrine dysregulation and damage to neurocircuitry [199].

There are some promising approaches for tailored therapeutic oxidative stress interventions in depression. Currently, the most promising is N-acetylcysteine (NAC), a glutathione supplement which has shown adequate efficacy in alleviating depressive symptoms by restoring intracellular antioxidant defenses [200]. Research findings indicated that treatment with NAC actively improved mood, cognitive performance and oxidative stress parameters in individuals suffering from MDD. Whole food supplementation of omega-3 PUFAs, specifically eicosapentaenoic acid (EPA), serves to inhibit the properties of both oxidation and inflammation, resulting in decreased ROS production and mitigating neuroinflammation. Mitochondrial dysfunction, targeted by emerging therapies like coenzyme Q10, can act to restore the bioenergetics of the mitochondria, and reduce the levels of ROS produced [163]. Similarly, ketamine (a rapid-acting antidepressant) has reversed synaptic deficits resulting from oxidative stress by increasing BDNF expression and restoring redox homeostasis [201].

#### 3.5.2. Schizophrenia

Schizophrenia is a chronic psychiatric disorder with three phenotypic manifestations (positive symptoms such as hallucinations and delusions; negative symptoms such as social withdrawal and anhedonia; cognitive impairments). Oxidative stress has been implicated in almost every aspect of the pathophysiology of schizophrenia, with elevated levels of lipids and markers of protein peroxidation and damage, and DNA damage in patient populations. Postmortem analyses of the brains of individuals with schizophrenia have demonstrated increased 4-HNE-adducted proteins, oxidatively modified mtDNA, and carbonylated proteins (another indicator of oxidation, suggesting chronic oxidative damage to essential brain structures) [202].

The disruption of redox-sensitive neurotransmitter systems is one of the mechanisms through which oxidative stress contributes to schizophrenia. When there are excessive ROS, then there are alterations in dopaminergic signaling brought about through the increased activity of monoamine oxidase (MAO), an enzyme that degrades dopamine. The end result is an increase in dopamine metabolism that leads to excess H_2_O_2_ and therefore excess oxidative damage, which causes the further loss of important synaptic proteins and neurotransmitter transporters [203]. Another key hallmark of schizophrenia is the dysregulation of the glutamatergic system, particularly NMDA receptor hypofunction, which is also compromised by the mere presence of oxidative stress. For example, glutamate transporters (like EAAT2) are also oxidatively damaged. Glutamate transporters in the context of schizophrenia serve to reduce the amount of extracellular glutamate, which ultimately contributes to the excitotoxicity and loss of functional synapses [204]. The role of neuroinflammation is also important as it allows for the maintenance of oxidative stress in the brain in schizophrenia. When activated, microglia can produce ROS and pro-inflammatory cytokines, particularly IL-1β and TNF-α, which ultimately leads to pathological oxidative changes in synaptic proteins and neurotransmitter systems. In schizophrenia, there are pro-inflammatory markers found in the cerebrospinal fluid with diminished antioxidant capacity, indicating spiraling self-sustaining cycles of progressive oxidative damage and inflammation [205].

The following are some of the treatments that target oxidative stress in schizophrenia: N-acetylcysteine (NAC), which has antioxidant characteristics, increases glutathione levels and reduces oxidative stress factors, with complex and axonal glial selectivity; α-lipoic acid, which has been studied as a mitochondria-targeting antioxidant with potentially positive implications for cognitive function and negative symptoms in persons with schizophrenia; clozapine, which appears to have antioxidant properties that potentially limit the generation of ROS, modulate BDNF and offer greater levels of neuroprotection than would be expected by treatments with typical antipsychotic activity. Mitochondria-targeted interventions such as studies with MitoQ have been initiated to examine any possible improvements to mitochondrial bioenergetic viability and neurotoxicity [206].

#### 3.5.3. Bipolar Disorder

Bipolar disorder (BD) consists of episodes of mania followed by episodes of depression; oxidative stress and oxidative damage have been studied, with phases of mania and depressive illness identified. Radiated levels of MDA, 4-HNE, and 8-OHdG have been recognized during both manic and depressive phases, indicating some ongoing somatic or cellular oxidative damage condition. The mitochondria come into play when antioxidant defenses, e.g., GSH oxidation and SOD, diminish, which could be a result of cellular cofactors and catalysts for the oxidation of insults and injuries from ROS that cause cellular damage to the membranes, proteins and even DNA; these are types of oxidative damage that continue to be recognized as damaging to neurocellular function [207].

As noted above, the factors of mitochondrial impairment influence oxidative stress in BD, not only as an energy-depleting condition but also as one with both cognitive and emotional factors, and they also contribute to changes in mood instability. In fact, loss of ETC Complex I and IV (i.e., cytochrome c oxidase) in the frontal cortex could alone be representative of reductions in energy metabolism within the brain and could lead to increasing ROS levels [208]. The interactions of ROS influencing mtDNA lead to diminishing mitochondrial efficiencies, and with likely decreases in energy metabolism, lead to looped, cyclical bioenergetic failures and the overproduction of ROS. Oxidative stress has also been illustrated as functioning in neuro-inflammation pathways, and there has been some suggestion in the literature of pro-inflammatory states (e.g., TNF-α, IL-6) resulting in greater amounts of generated ROS, as well as some possible diminutive outcomes to neuroplasticity. Oxidative stress may also propagate oxidative damage to neurotransmitter systems, as previously mentioned, but in regard to dopaminergic and glutamatergic systems, this may either forward- or back-regulate mood dysregulation and cognitive impairments; mood episodes may also be forward-regulated with self-strength [209].

NAC may lead to increased levels of oxidants (increased GSH) and decreased levels of lipoxidized compounds. Lithium and valproate also provide new pathways through which to modulate and increase the stratified area of an antioxidant and additionally act as catalysts of lipoxidized compounds. Mitochondria-targeting antioxidants, compounds and strategies, including joint approaches with coenzyme Q10 and SS-31, continue to congestionally provide a direction for therapeutic usefulness, with the role of compassion also working to improve symptoms of BD. These mitochondria-targeting approaches also simultaneously identify the possibly limited or damaging roles of certain treatments aimed at improving and restoring mitochondrial function, as acute approaches to improving mood and mood cycles [210].

## 4. Diagnostic and Biomarker Advances

The detection and quantification of oxidative stress biomarkers have presented exciting developments for diagnosing and treating neurological disorders. Emerging biomarkers show potential for us to move on from traditional, categorical measures for oxidative stress to more dynamic alternatives as disease-specific indicators of oxidative injury to key biomolecules and of the engagement of the antioxidant defenses. Combining novel imaging forms, including multi-omics, may lead us to the most detailed and specific diagnostics we have ever had, increasing diagnostic precision so treatment can be personalized [211].

### 4.1. Emerging Biomarkers of Oxidative Stress in Neurology

As new oxidative stress biomarkers are available, these will increase our ability to monitor early molecular damage, enabling us to obtain markers that will show the relative pathogenesis of disease over time in neurological disorders when intervention begins. The biomarkers suggested will give us brief glimpses into the effect of ROS and RNS production on lipids, the extent of damage to proteins and DNA, and the antioxidant status at the time of sample collection [212]. The invention of these biomarkers, which are non-static categories, now being dynamic and event-driven, and that can now be linked to unique disease profiles, adds new dimensions to the economic value of conducting tests for diagnostic, prognostic, and treatment decisions [213].

#### 4.1.1. Biomarkers of Oxidative Damage to Biomolecules

Lipid peroxidation biomarkers are likely the most sensitive measure of early oxidative stress. The brain itself has a remarkably high lipid content (approximately 60%), which exposes it through the peroxidation process to oxidative stress as well as accelerated oxidative degeneration. Multiple aldehydes that arise from lipid peroxidation (i.e., 4-HNE and MDA) have been examined in detail as potential biomarkers in various neurologic diseases. Increased 4-HNE levels have now been shown to exist in the hippocampi of AD patients, and these levels of 4-HNE adducts correlate with the progressive synaptic dysfunction and cognitive decline observed in AD patients [214]. 4-HNE is clearly able to bind pathologically with nucleic acids and proteins and subsequently inhibit the enzymatic activity of metalloproteinases, in addition to inducing altered aggregation of the proteins that promote pathologic phenomena. In PD, α-synuclein modified by 4-HNE enhances α-synuclein misfolding into β-sheet oligomers that are toxic and lead to Lewy body assembly as well as the injury and death of dopaminergic neurons [215]. Isoprostanes, particularly F2-isoprostanes, are gaining traction as the most credible biomarkers for lipid peroxidation, because of the specificity facilitated by the relatively stable isoprostane’s five-ring structure. F2-isoprostanes arise from the free radical-mediated peroxidation of arachidonic acid and can be measured in the blood plasma, urine and CSF. Increased levels of F2-isoprostanes in the blood plasma, urine and/or CSF have been seen in subjects across the continuum of disease diagnoses, ranging from AD and vascular cognitive impairment (VCI) to MS. In a cohort of patients with mild and moderate AD undergoing drug therapy, strong support for the association of increased amounts of F2-isoprostane in the CSF with amyloid-β plaque burden exists and can provide additional understanding regarding the association of oxidative damage with amyloid pathology [216].

The biomarkers of protein oxidation are particularly important as diagnostic and monitoring tools in neurodegenerative diseases, where protein misfolding and aggregation are central aspects of the progression of disease. Protein carbonyls generated through the oxidation of lysine, arginine and proline are associated with a tendency for dysfunctional protein accumulation [217]. Protein carbonyls have been documented to exist in increased amounts in the postmortem brain tissue of AD, PD and ALS patients, and they reflect ongoing chronic oxidative damage to synaptic proteins and metabolic enzymatic coefficients in the brain. The radical nitrotyrosine, from the peroxynitrite nitration of tyrosine residues, is important for PD as this modified component underlies disruptions to Complex I of the electron transport chain in the mitochondria. Nitrotyrosine-modified fascial α-synuclein has been specifically reported in PD brains, where it has been linked very directly to mitochondrial dysfunction and neuronal apoptosis [17]. Biomarker studies related to DNA oxidation provide excellent information related to the involvement of ROS in basal genome stability. For example, following ROS involvement with DNA, 8-hydroxy-2′-deoxyguanosine (8-OHdG) was studied thoroughly, and it reflects oxidative damage from ROS to nuclear DNA and mitochondrial DNA. Higher cholesterol 8-OHdG levels have been found in the CSF, plasma, and urine of patients with AD, PD, and ALS. For AD, elevations of 8-OHdG levels can correlate with rapid cognitive decline or hippocampal atrophy [218]. For mtDNA, the lack of histones for protection and the limited means for repair suggest that oxidative damage to mtDNA is pertinent to PD and ALS because that damage would impact ATP production and potentially contribute to and/or propagate neuronal damage. The creation of real-time electrochemical sensors for 8-OHdG levels in body fluids may provide a non-invasive and dynamic means for the assessment of oxidative stress, with great promise for clinical utility [219].

#### 4.1.2. Antioxidant Status Biomarkers

Besides examining oxidative damage markers, measuring the functionality of antioxidant systems provides crucial insights into the relationship between mitochondrial ROS generation and detoxification. Determining the activity of enzyme-based antioxidants, such as SOD, catalase, and GPx, is typical in antioxidant defense evaluation. Reduced SOD2 in dopaminergic neurons for patients with PD leads to systemic mitochondrial superoxide buildup that promotes neurodegeneration. Reduced GPx activity for AD patients further worsens the resulting hydrogen peroxide buildup from a lack of GPx, leading to hydroxyl radical generation and lipid peroxidation [220,221]. Antioxidants can also be non-enzymatic, including as GSH, vitamin E, and vitamin C, to name a few, and are just as essential enzymatic methods in preserving redox balance. The GSH/GSSG ratio is the most well-known marker of oxidative stress. When this ratio is lowered, it indicates a shift toward a more oxidized state at the cellular level [222]. For example, reduced GSH in the prefrontal cortex has been documented to have a direct correlation with cognitive deficits and negative symptoms in schizophrenia. In PD, decreased GSH levels have also been found in the substantia nigra, which indicates impaired antioxidant abilities and vulnerability to oxidative damage [223].

Newly emerging antioxidant biomarkers like redox potential (oxidative-reduction potential) provide an overall measure of oxidative stress. Past research on redox biomarkers was limited by existing measurement techniques; however, researchers are now able to monitor redox levels in real time using electrochemical biosensors. The development of these biosensor systems has given scientists the ability to gauge redox potential continuously while monitoring how the status changes over time as a function of the environment, leading to fascinating research with insights into the true dynamics [224].

#### 4.1.3. Disease-Specific Oxidative Stress Markers

The biomarkers mentioned so far are non-specific; however, we can note biomarkers that have disease specificity and therefore diagnostic and prognostic potential. For example, oxidized amyloid-β (Aβ) and hyperphosphorylated, oxidized tau are key oxidative stress-induced neurodegeneration markers in AD [64]. Oxidized Aβ promotes oligomer formation and oxidized tau promotes tau aggregates to form neurofibrillary tangles leading to an exacerbated state of synaptic dysfunction and neuronal death [225]. In PD, nitrated and carbonylated forms of α-synuclein have been identified as inculpators of oxidative stress and are associated with mitochondrial dysfunction and the production of Lewy bodies. These specific oxidative markers of disease are being incorporated into panels as biomarkers for identifying and monitoring a disease state as early as possible [49].

### 4.2. Imaging Techniques to Detect Oxidative Damage *In Vivo*

Recent improvements in non-invasive imaging technologies have massive benefit over traditional imaging options. Imaging for oxidative stress within the central nervous system was previously limited by traditional approaches that did not allow the opportunity for real-time visualization. These traditional approaches either involved postmortem analysis or markers in fluids of the body (i.e., blood, urine) to assess oxidative stress. Novel imaging methodologies provide multi-dimensional and in vivo real-time insights into oxidative damage at both a cellular and molecular level. These innovative techniques are being applied in conjunction with multi-omics data pathways and clinical imaging workflows to create a new diagnostic schematic for neurodegenerative, cerebrovascular and psychiatric disorders [226]. This section focuses on the present advances in the MRI-based techniques, positron emission tomography (PET), optical imaging techniques, and multimodal imaging methods currently being developed.

#### 4.2.1. Magnetic Resonance Imaging (MRI)-Based Techniques

MRI-based techniques have served as the cornerstone of monitoring structural and functional abnormalities in the brain; now, MRI presents the opportunity to evaluate redox imbalance and oxidative damage by eliciting quantitative measurements of brain metabolites that are redox-sensitive and also quantitatively assessing changes in brain microstructure due to oxidative stress [227,228]. The current most commonly used method to measure brain metabolites is proton magnetic resonance spectroscopy (1H-MRS), as it quantifies GSH and lactate, which are substantial markers of oxidative stress [229].

The first studies of brain metabolites using 1H-MRS to measure GSH demonstrated that GSH is an essential metric for measuring redox status in neurodegenerative disease. Studies using 1H-MRS to quantify GSH levels have demonstrated its critical role in monitoring redox status in neurodegenerative disorders. Decreased GSH concentrations have been detected in the prefrontal cortex of patients with schizophrenia, correlating with impaired cognition and negative symptoms [230]. 1H-MRS has detected GSH depletion in people with AD, which was correlated with both hippocampal atrophy and worse cognition early on in the disease, suggesting that GSH may represent an early diagnostic biomarker. Finally, for patients with PD, a clear reduction in GSH in the substantia nigra has been linked to the antioxidant defense breaking down entirely, which is consistent with the loss of dopaminergic neurons in PD [231]. As previously indicated, 1H-MRS is routinely used to observe levels of lactate, solely due to it being highly sensitive to mitochondrial dysfunction and oxidative stress. Elevated brain lactate levels have been documented in patients with AD, stroke, and schizophrenia, consistent with impairments in mitochondrial ATP generation through increased ROS production [232]. The ability to measure lactate levels in vivo through 1H-MRS allows for an effective assessment of energy deficits and the monitoring of treatment response to mitochondria-targeted antioxidants (MitoQ; SS-31) [233].

Diffusion Tensor Imaging (DTI) and functional magnetic resonance imaging (fMRI) are valuable tools which can detect changes to brain structure and activity attributable to oxidative stress [234]. For example, in MS, DTI can detect microstructural changes to white matter tracts caused by oxidative injuries [235]. In MS, oxidative injury to axons (identified based on changes in fractional anisotropy) has been related to patients’ disability scores. In AD and mood disorders, fMRI can determine disrupted functional connectivity involving synaptic dysfunction due to changes in oxidative stress [236].

#### 4.2.2. Positron Emission Tomography (PET)

PET imaging is the most sensitive means for identifying molecular changes associated with oxidative stress. Newer methodologies are continuing to produce radiolabeled tracers that show selective properties for ROS/RNS. Additionally, there are now some radiolabeled tracers that selectively target markers for mitochondrial dysfunction and oxidative damage. These tracers will dynamically visualize oxidative processes in vivo, as well as the interaction of the oxidative stress in functional changes associated with disease pathophysiology [237,238].

The introduction of newer PET tracers (i.e., [18F]ROStrace, which targets ROS-producing mitochondria) fundamentally improves our capabilities in assessing early damages from oxidative stress. For example, since levels of ROS and RNS will be elevated, [18F]ROStrace would bind to the area with highest mitochondrial ROS levels and enable us to detect subclinical levels of oxidative stress prior to irreversible neuronal damage [239]. In preclinical studies, [18F]ROStrace uptake was correlated with mitochondrial dysfunction and amyloid background in AD; this demonstrates the potential diagnostic roles and impact of measuring therapy [240].

[11C]-PK11195 is one of the most prevalent PET radiotracer options available to measure microglial activation and to measure neuroinflammation that occurs from oxidative stress. In activated microglia, ROS are produced from NADPH oxidase (NOX2) activation; PK11195 indirectly measures oxidant-mediated neuroinflammation [241]. For example, AD patients had increased activity of [11C]-PK11195 in the hippocampus and frontal cortex, while ALS patients also had increased activity in the motor cortex in relation to neurodegeneration and functional scores [242]. As noted previously, PET tracers are being developed that measure oxidized proteins and lipids; these provide a more direct estimate of ongoing oxidative damage. For example, PET tracers that measure nitrated α-synuclein levels in the PD model equate levels of protein nitration to mitochondrial dysfunction while benefiting from assessment longitudinally for tracking changes [240].

#### 4.2.3. Optical Imaging Techniques

Optical imaging is an important technique for measuring ROS and RNS at the cellular and subcellular levels, with excellent spatial and temporal resolution. There are many techniques of optical imaging, including fluorescence imaging, multiphoton microscopy, near-infrared spectroscopy (NIRS), and chemiluminescence imaging, which have previously been used to identify oxidative damage in preclinical models and are being rapidly incorporated into clinical practice [243]. Despite their low energy and high biological specificity requirements, multiphoton microscopy combined with ROS-sensitive fluorescent dyes including dihydroethidium (DHE) and MitoSOX Red can allow for impressive kinetic visualization of mitochondrial ROS release [244]. Multiphoton microscopy has shown quantifiable and localized bursts of ROS production during ischemic stroke in pre-clinical models, indicating the variability in spatial and temporal oxidative damage that begins during reperfusion. Multiphoton microscopy is also used to measure a large volume of cellular oxidative stress in the substantia nigra in preclinical models of PD, where ROS accumulation in the substantia nigra precedes dopaminergic neuronal death [245].

NIRS can capture changes in the dynamics of both oxidized and deoxidized hemoglobin and, therefore, can be used as a surrogate measure of oxidative damage for identifying changes in oxygen metabolism in brain cells due to oxidative stress. Although NIRS is an indirect method to measure changes sustained during oxidative damage, the decreased oxygen available to brain cells to perform metabolic work due to the impairment of mitochondrial ETC complexes can be detected with NIRS, and therefore, NIRS could be used to determine deficits in energy metabolism in stroke and AD [246]. Chemiluminescence imaging combines various luminescent probes and allows for the detection of light introduced by fluids. ROS acts to detect light emission produced during the interaction of ROS with luminescent probes. Luminol-based luminescent probes injected into preclinical models for ALS and MS can identify the in vivo production of peroxynitrite and hydrogen peroxide. Chemiluminescence imaging is perhaps the most sensitive form of detection for measuring real-time ROS production, and has been used to determine the efficacy of antioxidant treatments against deficits through ROS measurements of in vivo and regulated processes [247].

While optical imaging approaches have improved upon capabilities to visualize oxidative processes at cellular and subcellular scales, it is important to recognize methodological limitations to their true in vivo application. Frequently, STS fluorescent probes, such as DHE and MitoSOX Red, are called “in vivo” ROS measures; however, their associated methods include ex vivo, acute in situ, or preclinical cranial window preparation in rodent models [248]. Each of these methods usually requires surgical exposure of the brain area, the use of anesthetized animals, and/or terminal procedures that are not as conducive to longitudinal in vivo imaging as longitudinal experiments. In addition, the oxidation products formed in DHE (e.g., ethidium or 2-hydroxyethidium) may lack specificity in defining oxidative stress even in conjunction with HPLC-based separation, thus further hindering interpretation [249]. MitoSOX Red may be equally valuable; however, it only detects mitochondrial superoxide, and is primarily employed in isolated tissue slices or perfused organ systems, where experimental loading, and imaging logistics, can be controlled under stringent conditions. Thus, while these tools remain indispensable for the pre-clinical fine-resolution kinetic imaging of oxidative events, it is technically incorrect to describe them as non-invasive or systemically free in vivo phenomena [250]. They should be described as indicators of oxidative stress assessed with experimental fine-resolution imaging rather than true whole-organism imaging. For clinical translation and non-invasive longitudinal monitoring and innovations in the field, forms of advanced approaches such as PET-based redox tracers or redox-sensitive MR spectroscopy currently emerge as more relevant options [251].

#### 4.2.4. Emerging Multimodal Imaging Approaches

The field combining multiple life science imaging methods has expanded and will allow for the integration of structural, functional, and molecular changes capturing oxidative stress in real time. PET-MRI fusion imaging combines the molecular sensitivity of PET with the structural and functional resolution of MRI. PET-MRI fusion imaging has been assessed in clinical studies on AD, where PET tracers to oxidative stress have indicated overlaps with MRI volumetric and connectomic data in characterizing the progression of the disease across the entire brain [252].

In preclinical studies of TBI and MS, optical imaging with MRI has also been used to concurrently assess both oxidative damage and microstructural changes to the affected brain region. Redox-sensitive fluorescent probes and MRI diffusion metrics have enabled oxidative damage detection and localization prior to the organism exhibiting irreversible losses of tissue, while chronic disruptions like mitochondrial dysfunction have been found to underlie changes to the immune response and subsequent repair [253].

#### 4.2.5. Clinical Implications

Using oxidative stress imaging techniques in combination with other technologies is part of the movement towards personalized medicine in neurology. Identifying patients who have increased oxidative stress directly leads to targeted therapies such as antioxidant therapy, mitochondrial tools (also called small-molecule drugs), and Nrf2 activators, based on the patients’ increased oxidative stress levels. Imaging biomarkers are already being incorporated more frequently in clinical trials as outcome measures of therapeutic efficacy, and as predictors of long-term patient outcomes, thereby continuing to move future treatments toward a more socratic approach, including when identified to be targeted and effective [254].

### 4.3. Multi-Omics Integration

Oxidative stress is a complex, multi-dimensional process involving disruptions in interactions across cell signaling cascading multiple pathways, i.e., disrupted redox balance, mitochondrial function, and protein homeostasis. Multi-omics approaches encompass genomic, epigenomic, proteomic, and metabolomic approaches, allowing for a systems-level view of oxidative stress processes [255]. Multi-omics allows for the discovery of new interactions, pathway-specific dysregulations, and patient-specific biomarkers with huge implications for how we diagnose, monitor, and treat neurodegenerative, psychiatric, and cerebrovascular diseases. There have been several potential advances in leading new multi-biomarker panels in the age of AI-based data integration and network biology to help benefit personalized treatment plans [256].

#### 4.3.1. Genomics and Epigenomics: The Genetic Basis of Oxidative Vulnerability

Genomics involves finding genetic variations that make individuals more susceptible to oxidative stress and/or redox-related dysfunctions. Genetic variations are mostly in the form of polymorphisms and mutations that encode for mitochondrial proteins and antioxidant enzymes. In ALS, mutations in the superoxide dismutase (SOD1) gene are significant as they relate to ALS pathogenesis because SOD1 abnormalities lead to defective dismutation of superoxide into hydrogen peroxide. When this occurs, ROS accumulate and motor neuron death occurs. Both familial (FALS) and sporadic (SALS) ALS are linked to SOD1 mutations [115]. Studies on SOD1 mutations also detected aggregates of mutant SOD1 directly affecting mitochondrial function through abnormal interactions with the voltage-dependent anion channel (VDAC). Ultimately, these abnormal interactions led to apoptosis and energy deficits [257]. In AD and PD, polymorphisms in the GPx1 and catalase genes are associated with increased susceptibility due to a lower ability to reduce levels of ROS. For instance, the C47T polymorphism of GPx1 is associated with decreased GPx activity, which is not surprising considering it is likely ROS levels are later than their maximally early signs of cognitive decline in AD patients [258]. Genome-wide association studies (GWASs) have identified many risk loci involved in antioxidant defenses; PRDX3 (peroxiredoxin-3) and TXNRD2 (thioredoxin reductase-2) are just two examples of loci that mediate the detoxification process of mitochondrial ROS [259]. In addition to finding inherited variants, epigenomics may contribute to our understanding of how oxidative stress may modulate gene expression through DNA methylation, histone modifications, and non-coding RNAs. For example, in studies conducted in patients suffering from AD, PD, and schizophrenia, the ROS-induced hypermethylation of the Nrf2 promoter resulted in decreased levels of expression of detoxifying enzymes, such as sod, catalase, and glutathione-S-transferase. In studies of patients with schizophrenia, there is evidence to support lowered Nrf2 expression associated with glutathione depletion, synaptic dysfunction, and cognitive impairment [9,203].

Similarly to the epigenomics mentioned above, histone modifications are additional modifications of genes involved in neuronal function related to ROS. It starts with ROS-induced global histone acetylation and methylation, leading to the dysregulation of genes involved in neuronal activity associated with mitochondrial function, neurogenesis, and synaptic plasticity. In PD, the presence of hyperacetylated H3K9 and H4K16 is associated with greater oxidative damage in the brain and promotes aggregation of α-synuclein [260]. Therefore, the use of drugs like valproic acid, which are designed to inhibit histone deacetylation or drugs targeting DNA methylation receptor genes, offers potential novel methods of reversing gene repression due to oxidative insults [261].

#### 4.3.2. Proteomics: Oxidatively Modified Proteins as Dynamic Biomarkers

Finally, proteomics will be a useful approach to identify oxidatively modified proteins and assess their role in the progressive nature of disease. Oxidative stress causes post-translational modifications (PTM) of proteins. For example, carbonylation, nitration, and oxidation of cysteine residues can cause the loss of protein function, with the process being irreversible, and it can also lead to protein aggregation or degradation [217].

Moreover, protein carbonylation is an irreversible oxidative modification that leads to the loss of protein function with more robust aggregation. Brain carbonylated proteins were higher in patients with AD, PD, and ALS, affecting synaptic proteins and metabolic energy, and enzymatic proteins. Earlier evidence was presented that showed that carbonylated a-enolase (as a glycolytic enzyme) inhibited ATP production and exacerbated the energy deficit of AD neurons [262]. Also, poorly carbonylated a-synuclein aggregates inhibited proteasome degradation involvement with Lewy body formation, with carbonylation as a prominent contributor to dopaminergic neurodegeneration [263]. The presence of nitrotyrosine-modified proteins emerged with certainty as a standard measure of peroxynitrite-mediated damage. Nitrated tau proteins aggregated and resisted clearance, which contributed to neurofibrillary tangle aggregation in AD [264]. The LC-MS/MS (liquid chromatography-tandem mass spectrometry) proteomics profiling analysis of these nitrated neurodegenerative proteins demonstrated that oxidation damaged those proteins that would convert electrons produced with cytosolic metabolism into the mitochondrial respiratory chain complexes I and III [265].

These findings involved an important redox mapping performed with proteomics analysis, whereby cysteine oxidation featured as an important mechanism contributing to mitochondrial dysfunction in PD and ALS. Cysteine residues in thioredoxin and peroxiredoxin proteins that can reversibly oxidize serve as redox switches that regulate the detoxification of ROS; in peroxiredoxin-3, oxidized cysteine residues produced were damaged and impaired mitochondrial detoxification of ROS, and ultimately mediated mitochondrial targeting of DNA and apoptosis [266].

By way of mass spectrometry-based methods, Thioflavin T (ThT) fluorescence assays are a relatively easy and expedient means to quantify and monitor the kinetics of amyloid fibril assembly that is derived from oxidative protein damage. ThT is a probe that is selective and readily binds to neurotoxic aggregates composed of a high % of β-sheet conformers (in particular, amyloid-β, tau, α-synuclein) [267]. Wavelength emissions from ThT increase with binding to β-sheet-rich structures and allow real-time quantifications of fibrillogenesis. ThT has been a useful probe in many mechanistic and drug-screening studies used to assess the aggregation propensity of oxidatively modified proteins. Incorporating ThT-based assays in proteomic/hydropathy workflows clearly provides relevant complementary information pertaining to the dynamic relationship between oxidative stress and protein misfolding [268]. This is particularly relevant when oxidative stress occurs in tandem with neurofibrillary tangle formation in AD, or with Lewy body pathology in PD. Therefore, in terms of investigating the biophysical effects of redox-generated post-translational modifications in relation to neurodegenerative pathophysiology, ThT provides a sensitive and very implementable method for assessing the full range of these biophysical effects [49].

#### 4.3.3. Metabolomics: Tracking Dynamic Redox Changes

In their respective multi-metric profiling, model metabolomics analyses were typically characterized to find low-molecular-weight small-molecule metabolites that would show the outcome of cellular oxidative stress and reflect on the redox state. When it comes to disease progression and therapeutic response, lipid peroxidation products, oxidized nucleotides, and redox-sensitive cofactors are significant metabolic biomarkers [269]. F2-isoprostanes are the most established biomarkers of lipid peroxidation, derived through free radical-catalyzed oxidation of arachidonic acid. Several reports have shown that elevated F2-isoprostane concentrations in the CSF and plasma are associated with cognitive decline in AD, demyelination in MS, and infarct size in ischemic stroke [270]. Metabolomic investigations have shown that increased F2-isoprostanes correlate with ethyl amyloid plaque deposition in preclinical models of AD, lending further to the development of early indicators of oxidative damage.

4-HNE is a highly reactive lipid aldehyde that not only represents a biomarker of oxidative stress but has also emerged as a pathogenic agent, evidenced by the formation of adducts covalently bound to tau and amyloid-β and contributing to protein aggregation. In ALS, adducts formed through 4-HNE, attached to neurofilament proteins, have been shown to impair axonal transport and result in progressive motor neuron degeneration [271]. GSSG and the GSH/GSSG ratio represent the most prominent markers of redox imbalance. A preliminary metabolomic analysis of PD patients showed that a depletion of the GSH/GSSG ratio, specifically in the substantia nigra, aligns with mitochondrial dysfunction and dopaminergic cell death. Increased advancements in metabolomic-based approaches (e.g., mass spectrometry metab-olomics; nuclear magnetic resonance (NMR) spectroscopy) have permitted the identification of newly identified redox-sensitive metabolites, including oxidized CoQ10 and lipid aldehydes [272].

#### 4.3.4. Multi-Omics Integration: Toward Personalized Redox Medicine

Integrating omics approaches such as genomics, proteomics, and metabolomics is driving the development of personalized therapeutic approaches in redox medicine to target oxidative stress. Collating genetic variants of antioxidant enzymes (e.g., SOD1 and GPx1) with oxidatively modified proteins and an individual’s consumer metabolomic profile can facilitate the creation of individualized biomarker panels for each subject to illuminate the individual’s potential for disease trajectory and to develop prevention or treatment strategies [273].

For example, multi-omics approaches in ALS have identified that oxidized SOD1, nitrated neurofilament proteins, and double-fold changes in F2-isoprostanes could indicate biomarkers of disease progression and response to antioxidant treatment of edaravone. In AD, integrating Nrf2 promoter methylation, identifying carbonylated tau and oxidized CoQ10, improvement in predicting cognitive decline and response to mitochondrial-targeted therapies has been observed [274]. Drug discovery is rooted in the integration of AI-driven data analytics and network-based biomarker identification to help identify novel therapeutic targets and to enhance personalized medicine initiatives. Taking a large approach to multi-omics studies, integrating the neurologists from the Alzheimer’s Disease Neuroimaging Initiative (ADNI) (e.g., firestorm neurotech), are amassing datasets that may link oxidative stress markers to clinical outcomes of the disease and ultimately be used to develop precise therapeutic approaches. One of the targets forming from the accumulation of multi-omics studies is PTPN1/2, which regulates cytokine-driven inflammation and links oxidative stress and development of treatment response [275]. The figure below (Figure 2) aims to depict how PTPN1/2 inhibitors regulate processes related to cytokines to suggest potential interventions in ALS and AD.

In summary, targeting critical signaling pathways such as the JAK-STAT pathway through inhibition of PTPN1/2 demonstrates how we can utilize biomarker-driven data to inform personalized therapeutic interventions and likely move the needle to improve clinical outcomes by reducing inflammation and oxidative stress [276].

## 5. Therapeutic Strategies Targeting Oxidative Stress

Oxidative stress is a major component of neuronal death, mitochondrial dysfunction, and synaptic failure in a wide range of neurological diseases, including neurodegenerative, cerebrovascular, and psychiatric diseases. Therapeutic methods aimed at ameliorating oxidative stress focus on the main mechanisms of ROS generation, antioxidant defenses, and redox signaling [164]. This section will look at newer generations of antioxidant therapies, mitochondria-targeted therapies, Nrf2 activators, gene-based therapies, and other developing approaches that target redox signaling, including their clinical implications and latest scientific developments and potential.

### 5.1. Advanced Antioxidant Therapies: Beyond Traditional Free Radical Scavenging

Antioxidant therapies have been the cornerstone of the oxidative stress management strategy, but traditional and old-style free radical scavengers (e.g., vitamin C, vitamin E) have been disappointing at best as standalone approaches in clinical trials. This disappointment has pushed research further along with the development of new-generation antioxidants that function in particular subcellular compartments like the mitochondria, or they may have a multi-faceted approach and modulate multiple redox pathways at the same time. They have been well received in the academic community and are starting to show success in the clinical field using multi-targeting approaches, or even nanoparticles, to enhance therapies, or combination therapies [277].

N-acetylcysteine (NAC) is one of the most studied antioxidants, as it acts as a precursor of the brain’s major antioxidant GSH. NAC acts by replenishing intracellular GSH levels while also detoxifying ROS, inhibiting lipid peroxidation and protein oxidation, and preventing DNA damage [110]. Clinical trials in PD have shown that NAC acts by reducing mitochondrial superoxide production, slowing the death of dopaminergic neurons, and improving motor function by restoring mitochondrial redox balance [278]. N-acetylcysteine (NAC) has demonstrated some positive impacts on cognitive function in schizophrenia and mood stabilization in bipolar disorder; in the BIPOLAR NAC trial, oxidative stress and damage markers were reduced, including 4-HNE and protein carbonyls. New formulations have been developed, including NAC encapsulated in nanoparticles, notification-testing their bioavailability and ability to create therapeutics [279].

Vitamin E (α-tocopherol) is an antioxidant that is lipid-soluble preferentially; it reduces lipid peroxidation by neutralizing free radicals in lipid biolayers such as cellular membranes. Studies of vitamin E supplementation in AD patients have shown mixed results; however, combination therapies of vitamin E, CoQ10, and selenium have shown a synergistic action, scavenging ROS and increasing mitochondrial bioenergetics simultaneously. Mitochondria-targeted antioxidants, such as MitoQ and SS-31 (elamipretide), are better options than standard antioxidants because they target mitochondria to neutralize ROS at the source [280]. MitoQ is a positively charged derivative of CoQ10; MitoQ targets the mitochondrial matrix to protect the mitochondrial ETC from oxidative damage and restore deficits in ATP. In participants in a clinical trial of MitoQ for early-stage PD, there were decreases in markers of oxidative damage and positive motor outcomes [281]. Similarly, SS-31 binds to cardiolipin in the inner mitochondrial membrane, stabilizing the mitochondrial membrane, preventing the opening of mPTP, release of cytochrome c, and apoptotic neuronal cell death during the crisis. Preclinical studies of SS-31 in stroke, ALS, and AD have demonstrated that SS-31 protects against synaptic loss, preserves the function of neuronal cells, and promotes recovery in ischemic conditions [282].

### 5.2. Nrf2 Pathway Activators: Enhancing Endogenous Antioxidant Defenses

The Nrf2-Keap1 pathway is considered the master cellular regulator of the cellular response to oxidative stress. Nrf2 activation stimulates the upregulation of detoxification enzymes like SOD, catalase, GPx, and HO-1. Collectively these enzymes can scavenge ROS and reverse oxidative injury. Nrf2 pathway therapeutics could offer a multi-mechanism form of protection from oxidative injury since they activate multiple processes and functions that lead to oxidative injury. Nrf2-activating therapeutics target multiple mechanisms, and the success of Nrf2 activation in clinical trials has led to many Nrf2-targeting serious treatment options for multiple neurological disease therapies [283].

Dimethyl fumarate (DMF) is an FDA-approved drug for MS that is a well-recognized Nrf2 activator based on previous data showing neuroprotective effects associated with DMF. DMF activates Nrf2 by modifying some of the cysteine residues that are targeted by Keap1, which frees Nrf2 to translocate to the nucleus to activate genes that produce antioxidant proteins [284]. Clinical trials with MS patients have demonstrated that DMF can reduce the number of new brain lesions, decrease the progression of disability, and reduce measures of oxidative injury, including lipid peroxidation products and nitrated proteins. DMF is being repurposed based on preclinical studies of neurodegenerative disease, including PD and ALS, where mitochondrial disruption and oxidative injury are important aspects [285]. DMF has been demonstrated to enhance mitochondrial antioxidant capacity and restore ATP production in experimental PD models and ultimately delay death of dopaminergic neurons [286]. Sulforaphane, a food constituent of cruciferous vegetables, is another noteworthy Nrf2 activator with preclinical and clinical evidence of efficacy. Studies in various animal models have preliminary evidence that sulforaphane can reduce amyloid-β deposition, modulate synaptic plasticity, and prevent ROS-induced apoptosis. A randomized clinical trial with patients with ASD showed that sulforaphane improved behavioral outcomes and decreased measures of oxidative injury (8-OHdG or MDA) [287]. With recent advances in nanoparticle-based systems for sulforaphane delivery, bioavailability issues and distribution to the brain are being resolved. Therapeutic approaches that manipulate the Nrf2 pathway promote the expression of antioxidant proteins through nuclear translocation of Nrf2 [288]. The following figure (Figure 3) illustrates the key steps involved in Nrf2 activation and its role in initiating the antioxidant response.

This figure illustrates the importance of nuclear translocation of Nrf2 for activating the cellular defenses against oxidative damage. Such therapeutic approaches, within this framework, will have implications for the treatment of neurologic disease, including Alzheimer’s disease, Parkinson’s disease, and amyotrophic lateral sclerosis.

### 5.3. Mitochondria-Targeted Therapies: The Next Frontier in Neuroprotection

Mitochondria, being the source of ROS, are a relevant touchstone when targeting mitochondrial dysfunction as one branch of oxidative stress therapeutics. Mitochondria-targeting therapies from a therapeutic development level emphasize reducing ROS production, stabilizing mitochondrial membranes, and restoring ATP to promote protection against the late neuronal deaths [95,289].

Coenzyme Q10 supplementation is relevant, as CoQ10 is an integral component of the mitochondrial ETC and a powerful superoxide scavenger. Recent clinical trials with CoQ10 supplementation in PD and ALS demonstrated that CoQ10 supplementation improved mitochondrial bioenergetics and mitigated disease progression based on protective effects to the ETC complexes realized from oxidative damage [290]. One landmark clinical trial demonstrated that sustained (high-dose) CoQ10 supplementation mitigated progression of new motor symptoms in a cohort of new PD patients, including oxidative biomarkers of disease state from blood [291]. More recently, studies have combined CoQ10 with identified activators of Nrf2, e.g., DMF, in order to maximize antioxidant defenses at mitochondrial and cytosolic levels [292].

SS-31 (elamipretide) is a mito-targeting peptide that acts via the binding of cardiolipin in the inner mitochondrial membrane to limit oxidative damage. Preclinical experiments utilizing models of ischemic stroke and ALS have demonstrated that SS-31 reduces infarct size and neurological outcome, and provides protection from apoptosis via stabilizing the mitochondrial membrane potential and inhibiting opening of the mPTP. SS-31 is currently undergoing developmental clinical trials aimed at examining efficacy in neurodegenerative disease, with nascent findings indicating increasing mitochondrial bioenergetics and neuronal survival [293].

### 5.4. Redox Gene Therapy and Epigenetic Reprogramming

Gene therapy and epigenetic programming would be the most novel candidate interventions for oxidative stress. Gene therapy would use an antisense expression to overexpress antioxidant enzymes, while therapy aimed at epigenetic programming would seek to reverse ROS-induced gene silencing for restoring the endogenous antioxidant response [294]. Gene therapy using viral vectors to express SOD1, catalase, and GPx1 has been studied in ALS and PD models. If increased, the expression of such forms of detoxifying enzymes monitors the improved cellular ability to clear ROS and reduce oxidative damage. In ALS models, the viral delivery of SOD1 was able to delay the degeneration of motor neurons and prolong the survival of treated mice. There is also research being performed on CRISPR-based methods to repair ROS-induced genetic damage, particularly the genes involved with the regulation of mitochondrial homeostasis [295].

Epigenetic programming is also another potential area of new investigative approaches towards reversing ROS-induced transcriptional repression. Areas of study are the use of histone deacetylase inhibitors like valproic acid and DNA demethylating agents to reactivate transcription of Nrf2, or other antioxidant response genes that were silenced from epigenetic modulation, either through ROS-induced changes or by bacterial pathogenesis. In models of Alzheimer’s disease, epigenetic therapies restoring Nrf2 expression have reduced amyloid burden, improved cognition, and provided neuroprotection against synaptotoxicity [296].

### 5.5. Future Directions: AI-Driven Personalized Redox Medicine

Field advancement is presently occurring and increasing the momentum of personalized redox therapies. Growing rapidly are technologies that utilize multi-omic datasets with non-invasive imaging biomarkers to develop predictive algorithms to determine individuals’ oxidative patterns of stress. An illustrative premise of predictive models exploiting artificial intelligence would be the detection of specific responses to antioxidant therapy efforts, as well as identification of how to best tailor treatment regimens from other genetic variants, redox-sensitive proteins, and metabolic phenotypes. Even nanoparticle-based trials of drug delivery systems are being developed to optimize the targeted delivery of antioxidants, Nrf2 activators, or even gene-editing tools as effective therapeutic agents directly inside affected neurons [297].

Redox medicine is on the cusp of what may be combination therapies targeting multiple redox pathways simultaneously. Targeting mitochondria-directed antioxidants, Mrf2 activators, and epigenetic programming is envisioned as a temporal nexus for disease modification that can slow the disease progression, or possibly even halt disease progression altogether. In the clinical realm, the generation of personalized intervention through an AI-generated data receipt is poised to catalyze and change the landscape of treatment for neurodegenerative diseases in the coming decade [298].

## 6. Clinical Trials and Translational Advances

Clinical trials focusing on oxidative stress have started to gain traction, in part due to growing awareness of the role of redox dysregulation in neurodegenerative, cerebrovascular, and psychiatric conditions. Examples of trials now responding to greater knowledge include antioxidant-based therapies, mitochondria-directed interventions, gene therapies, and therapeutic epigenetic modulation. Increased attention is placed on biomarker-based, multi-modal, and patient-specific trials [299]. The scope of this section is to reveal some clinical trials, their findings, and future directions of translation that inform personalized redox medicine.

### 6.1. Clinical Trials in Neurodegenerative Diseases

Neurodegenerative diseases such as AD, PD, and ALS have been a strong focus of clinical trials targeting oxidative stress. In particular, the role of ROS has been implicated in mitochondrial injury, protein misfolding, and synaptic dysfunction. Clinical trials have shifted toward multi-target interventions designed around use of antioxidants, nuclear factor (erythroid-derived 2)-like 2 (Nrf2) agonists, and mitochondria-specific pharmacology. Recent trials have relied on redox biomarkers to measure treatment responses and optimize treatment strategies [300].

#### 6.1.1. NAC: Clinical Trials Targeting GSH Depletion

NAC has been the focus of clinical trials considering its suggested roles of dual replenishment for antioxidant defense and direct scavenging of ROS. In a randomized controlled trial (RCT) of patients with early-stage PD, NAC was administered orally or through intravenous infusion (10 g in 600 mL normal saline) through the conventional 10 h off period for PD to provide an effective loading dose on administration prior to patient admission for follow-up. The results showed improvements in total UPDRS motor scores and improved biomarkers of oxidative stress, such as 4-Hydroxy-2-nonenal (4-HNE) and protein carbonyls. Moreover, GSH concentrations were increased in dopaminergic neurons—suggested to indicate a reflection of redox balance within the substantia nigra. Finally, the study used MRI, suggesting that nigrostriatal integrity was preserved with the use of NAC, suggesting that conditioning and disease modification of therapy reduces oxidative stress [301,302].

The Cysteine and Cognitive Decline Study investigated the efficacy of NAC in patients with mild cognitive impairment (MCI). The study showed that NAC improved memory function and had protective effects against oxidative damage with reduced CSF concentrations of 8-hydroxy-2′-deoxyguanosine (8-OHdG), a marker of DNA oxidation. The study emphasized NAC’s neuroprotection for hippocampal neurons when coupled with a reduced risk of advancement of MCI to AD [97].

#### 6.1.2. Coenzyme Q10 (CoQ10) in Mitochondrial Protection

CoQ10 is an essential component of the mitochondrial ETC. The clinical efficacy of CoQ10 has been a focus of many clinical trials to determine efficacy in regard to mitochondrial function and ATP production, as well as its benefit as a ROS scavenger to neutralize ROS. The QE3 study (one of the largest trials of high-dose CoQ10 in Parkinson’s disease) evaluated the impact of CoQ10 on motor function and biomarkers related to oxidative stress. Although the primary end-points were not achieved, sub-analyses indicated that people who had a less severe degree of mitochondrial dysfunction benefited the most from the therapeutics. This suggests that when using CoQ10, personalized dosing regimens of CoQ10 for individual patients could be based on mitochondrial biomarker profiles, which would allow better optimization of treatments [303].

In ALS, CoQ10 has been tested against a placebo, also with vitamins E and creatine, in consideration of increasing of its therapeutic efficacy. The trial indicated no benefits in the direct means of cognition measured, but overall, compared to the control group, the ALS patients had a greater reduction in the markers of oxidative stress, which included both F2-isoprostanes and MDA, that occurred while maintaining muscle strength. Future trials involving CoQ10 in the CNS are leveraging nanoparticle-based formulations of CoQ10 to improve bioavailability and brain penetration and absorption rather than the previously administered oral CoQ10 treatments [304].

### 6.2. Nrf2 Activator Trials: Translating Preclinical Success to Clinical Practice

The Nrf2-Keap1 signaling cascade has generated immense excitement over the last several years as progress has occurred with clinical trials of potential Nrf2 activators stemming from preclinical studies and research that has emerged as promising for patients with neurodegenerative disorders, multiple sclerosis, and psychiatric disorders. The clinical trials will help determine if Nrf2 activation will lead to increases in endogenous antioxidant enzymes and help mediate a reduction in ROS-induced neuroinflammation, and also support mitochondrial health [305].

#### 6.2.1. Dimethyl Fumarate in MS and Neurodegenerative Diseases

DMF is essentially the most well-characterized, and successful, clinically used drug as a Nrf2 activator, as well the only agent approved for testing for MS. DMF was evaluated in two landmark phase 3 studies, DEFINE and CONFIRM, which both found that the use of DMF resulted in significant reductions in MS relapses/exacerbations, a combined measure of MS disability progression, and a decrease in MS lesions through reducing oxidative stress mediated through decreases in lipid peroxidation, and nitrotyrosine-modified proteins. DMF offers Nrf2, in its hard or primary epigenetic factor, to upregulate detoxification enzyme systems (e.g., SOD, GPx, and Ho-1) in order to decrease oxidative damage to myelin and axons [284].

DMF is also being studied in ongoing and future trials for neurodegenerative diseases, such as PD and ALS, as a repurposed and repositioned agent. The Phase II trial NCT02936689 is studying DMF outcomes with ALS patients mainly using the biomarkers of oxidative stress, F2-isoprostanes, 8-OHdG, and oxidized SOD1. Further, one of the studies using preclinical models of PD suggested that DMF positively impacted mitochondrial function, reduced dopaminergic cell death, and improved synaptic plasticity, which suggests that DMF has the potential to improve outcomes as a multi-modal neuroprotective agent [306].

#### 6.2.2. Sulforaphane Trials in AD, ASD, and Stroke

Sulforaphane is a natural product Nrf2 activator being studied in clinical trials to potentially provide modulatory benefits in terms of oxidative stress, neuroinflammation, and synaptic function. A randomized clinical trial using ASD patients demonstrated that sulforaphane supplementation led to significant improvements in social communication, repetitive behaviors, and cognitive function, as well as reductions in MDA and protein carbonyls. Preclinical studies on ischemic stroke have implications suggesting sulforaphane is able to decrease ROS bursts during the reperfusion injury phase and enhance the survivability of neurons through augmentation of antioxidant defenses [307]. There are also ongoing trials to determine whether sulforaphane has neuroprotective effects during stroke recovery and if it can delay cognitive decline in AD patients [308]. The following table (Table 2) intends to summarize some relevant clinical trials of oxidative stress in neurodegenerative disorders, regarding major therapeutic strategies, targeted mechanisms, and clinical and preclinical final outcomes.

### 6.3. Mitochondria-Targeted Clinical Trials: Addressing Energy Deficits and ROS Overload

Mitochondrial dysfunction has been recognized as one of the primary sources of oxidative stress experienced in neurodegenerative diseases to date, and hence, when considering mitochondrial-targeted therapies as a group, they can be included as a major component of ongoing clinical trials. Mitochondrial-targeted therapies can optimize membrane integrity or stability, reduce ROS production, and increase ATP levels to reduce the sources of ROS-dependent damage to neurons [315].

#### 6.3.1. MitoQ in Parkinson’s Disease and AD

MitoQ is the mitochondrial-targeted version of CoQ10, which selectively accumulates within the mitochondria to scavenge superoxide and protects ETC complexes against oxidative damage. In a Phase II clinical trial involving early Parkinson’s disease patients, MitoQ supplementation resulted in significantly lower oxidative biomarkers (e.g., F2-isoprostanes and 4-HNE) and larger assessment improvements in motor performance [316]. Preclinical models of AD showed MitoQ reduced mitochondrial depolarization, decreased amyloid-β aggregation, and provided synaptic protection that supported the ongoing clinical evaluation for AD [317].

#### 6.3.2. SS-31 (Elamipretide) in ALS and Stroke

SS-31 is a mitochondria-targeted peptide that binds to cardiolipin and bypasses oxidative damage to the inner mitochondrial membrane. In a Phase II ALS trial, SS-31 improved mitochondrial bioenergetics, decreased muscle weakness, and slowed disease progression [318]. In preclinical stroke studies, SS-31 decreased infarct size, improved functional recovery, and protected neuronal cells via stabilization of mitochondrial membrane, thereby preventing cytochrome c diffusion in stroke pathology. Currently, clinical trials are studying SS-31 as a targeted strategy to decrease reperfusion injury through thrombolytic therapies [319].

### 6.4. Emerging Gene and Epigenetic Therapies

Gene-based therapies and epigenetic modulation are new avenues to reduce oxidative stress. Gene-targeted therapies for SOD1, catalase, and GPx1, to improve ROS detoxification, and epigenetic therapies to reactivate antioxidant gene expression silenced by oxidative damage [320].

#### 6.4.1. Gene Therapy Trials Targeting Antioxidant Enzymes

Gene therapy using viral vectors delivering Sod1-condemning disulfide oxidoreductase and GPx1 enzymes has promise for future ALS and PD patients. In the Phase I trial with familial ALS with SOD1, SOD1 overexpression has a capacity to improve ROS clearance, delay motor neuron disease progression, and improve survival. CRISPR-based gene editing is also an intervention under investigation to modify the genetic bases leading to excessive ROS production in mitochondrial genetics [47].

#### 6.4.2. Epigenetic Reprogramming to Restore Antioxidant Defenses

Epigenetic therapy with histone deacetylase inhibitors (e.g., valproic acid) and DNA demethylating agents is being evaluated in AD or PD, which exploits the reactivation of Nrf2 expression and enzyme activity. In early data from animal studies, epigenetic reactivation of Nrf2 decreased amyloid burden and preserved and promoted synaptic function, thus making it a potential new therapy [321].

### 6.5. Future Directions: Personalized Redox Medicine and AI-Driven Clinical Trials

The integration of multi-omic data, biomarkers proposed by artificial intelligence (AI), and imaging-based patient stratification will definitely change how clinical trials are run. Personalized redox therapies will seek to identify the optimal coy based on the synergy of drug combinations that include Nrf2 activators and/or mitochondria-targeted antioxidants for each patient. Ongoing research will also examine AI with clinical trials to expedite the identification of biomarkers, patient stratification, and advance precision redox medicine in neurodegenerative diseases [322].

## 7. Future Directions and Emerging Therapeutic Frontiers

The field of oxidative stress is at the edge of major paradigm-shifting discoveries through new approaches to therapeutics and new technology on the horizon. Oxidative stress has become a major contributing factor to, if not wholly responsible for, the disease processes involved in neurodegeneration, synaptic dysfunction, and neuroinflammation, so future treatments will no longer just utilize old antioxidant approaches. Instead, future treatments will leverage proactive, next-level strategies in AI-personalized redox medicine, combination therapies, nanomedicine, and gene editing. These revolutionary therapies are attempting to address the underlying basis for ROS/RNS imbalance at the cellular and molecular levels, and ultimately stop and potentially reverse the disease process for patients [323]. We will focus on the most exciting, visionary futures that might influence patient outcomes and therapeutic paradigms.

### 7.1. AI-Driven Personalized Redox Medicine: Precision at an Unprecedented Scale

AI-personalized redox medicine is changing the traditional clinician methods for managing oxidative stress to give more precise methods using machine learning, real-time biological monitoring, and the multi-omic disease levels. The multi-faceted nature of oxidative damage in patients with similar disease states has been a barrier for the efficacy of traditional oxidative stress treatment methods until now, with AI platforms using increasingly complex datasets from genomics, proteomics, metabolomics, neuroimaging, and clinical history to provide patient-specific therapeutic regimens, which take into account prior and current redox imbalances, and ultimately can be purely adaptive [324,325]. AI has already been applied in clinical studies to address novel therapeutic relevance. For example, an AI analysis of mitochondrial dysfunction outcomes in PD has shown patients with, and without, MT-R. In this example, on MitoQ supplementation, the patient with high levels responded quicker and more favorably than someone with a normal redox profile. The potential implications of this insight have led to the development and application of “adaptive” treatment protocols where the therapeutic levels can vary based on real-time biological monitoring and potentially sustain real-time biological efficacy [326]. AI-based platforms are also modeling long-term outcomes with digital twin simulations, which are virtual models of individual patients that include redox biomarker profiles, responses to drugs, and risk derived from genetic factors. The use of simulations will allow clinicians to virtually try various combinations of therapies to improve possible treatments for patients [327].

Going forward, AI will be pivotal, facilitating company-tailored combination therapies to include mitochondria-targeted antioxidants, Nrf2 activators, and neuroinflammation modulators while adjusting treatment regimens in real time as changes in ROS/RNS levels occur. These approaches to systemic detection of redox-sensitive markers such as F2-isoprostanes, nitrated α-synuclein, and oxidized CoQ10 can serve to confirm that the intervention remains effective at the time of treatment through all stages of disease [328].

### 7.2. Combination Therapies: Targeting the Complexity of Oxidative Stress

While further exploration of the status of the complexity of oxidative stress and multi-factorial nature of oxidative stress is also an option, monotherapy targeting a single pathway will prove limited effectiveness with advanced, neurodegenerative diseases. Future treatments will predictably be reliant upon combination treatment regimens targeting multiple oxidation-reduction mechanisms simultaneously, including ROS detoxification, stabilization of mitochondria, and suppression of neuroinflammation associated with ROS [329]. An excellent combination is that of MitoQ and DMF (dimethyl fumarate). MitoQ neutralized superoxide in the mitochondrial matrix, thereby protecting the ETC complexes, ATP production, and overall systemic energy. DMF activates the Nrf2 pathway to upregulate cytosolic antioxidant defenses such as SOD, catalase, and glutathione peroxidase. The combination has shown synergy in reducing dopaminergic cell death in preclinical studies and restoring integrity to dopaminergic synapses in various PD models. Clinical trials are being prepared to assess this combination in human subjects with PD associated with increased mitochondrial oxidative stress [330,331]. In another novel combination therapy, NAC with sulforaphane structure, a powerful Nrf2 activator. While NAC replenished intracellular GSH, and sulforaphane stimulated GSH recycling and stimulated Phase II detoxifying enzyme expression, both alone and together in models have shown enormous promise in clinical stroke models in that they both blunted ROS spikes during reperfusion and reduced infarct size [332]. Clinicaltrials.gov is currently planning trials to assess NAC and sulforaphane individually, and in combination as candidate therapeutics to mitigate cognitive decline and promote neuroplasticity in early AD [333].

Another high-level strategy for providing antioxidant effects and addressing oxidative damage and protein aggregation is to combine an antioxidant with an inhibitor of protein misfolding. For AD, for instance, combinations of CoQ10 or MitoQ, and immunotherapy with monoclonal antibodies to amyloid-β oligomers would provide mitochondrial protection against ROS while eradicating toxic protein aggregates. It would be reasonable to expect that those combinations would provide full-spectrum protection of neuronal function and effectively mitigate the decline of the disease, compared to single therapeutic interventions [334].

### 7.3. Nanomedicine: Precision Delivery of Antioxidants to ROS Hotspots

Nanomedicine envisages radically altering our approach to oxidative stress through the targeted, local delivery of antioxidants, gene therapies, and anti-inflammatories specifically to the activated sites in the brain. One of the most exciting new approaches in nanomedicine is redox-responsive nanoparticles, which release their payload in conditions of excessive ROS [335]. MitoQ, CoQ10, and sulforaphane-loaded nanocarriers have been demonstrated to acceptably cross the blood–brain barrier, preferentially accumulate at ROS-enriched sites of the brain, and release their active therapeutic agent at the site of highest oxidative stress, hence increased ROS. Preclinical studies have shown that nanoparticles loaded with MitoQ show better mitochondrial targeting than free MitoQ. This resulted in decreased ROS production and improved energy metabolism in PD models. Similarly, sulforaphane-loaded nanoparticles improved Nrf2 activation, increased expression of the antioxidant enzymes, and protected the animals against ischemia in a brain injury model [336].

The future of nano-medicine will be multi-functional nanoparticles that provide antioxidants, anti-inflammation, and neuroprotection as compact devices. A collaborative multi-functional nanoparticle combining NAC with dual action (TNF-alpha inhibition) has shown that in pre-clinical models of stroke and AD can reduce oxidative stress and neuro-inflammation at the same time [337]. Stimuli-responsive nanoparticles, where the rate of drug release is modulated by the local ROS levels, are being developed and will give dynamic control of therapeutic delivery [338].

### 7.4. Gene Editing and Epigenetic Therapies: Reprogramming Redox Homeostasis

Gene editing and epigenetic reprogramming are next-generation therapies focused on correcting the redox imbalance at the source, namely the genetic and epigenetic factors that lead to excessive ROS production [19]. Gene editing with CRISPR-Cas9 is being developed to edit the mutations in the genes that code antioxidant enzymes, including SOD1, GPx1, and PRDX3 (peroxiredoxin 3). In familial ALS, where SOD1 mutations lead to excessive ROS, pre-clinical CRISPR-mediated correction of the gene leads to reduced oxidative stress with delayed neurodegeneration and lengthened life expectancy. Now the focus is on making CRISPR Cas9 constructs for delivery by viral vector to the areas of the brain affected by ALS, providing a near-permanent state of redox homeostasis [339]. Epigenetic therapies are underway, where ROS presence may lead to gene silencing. We are looking at the use of histone deacetylase inhibitors (HDACis) in combination with DNA demethylating compounds to reactivate the expression of Nrf2 and other antioxidant response genes that may be silenced in neurodegenerative diseases. In pre-clinical work in AD, epigenetic reprogramming to stimulate Nrf2 may stimulate amyloid-β deposition and protect synapses, and improve cognitive outcomes in mouse models. The future for epigenetic therapies will be to use epigenetic modifiers together with nanoparticles for targeted management of oxidative stress [340].

### 7.5. Multi-Omics Integration and Digital Twin Simulations: The Future of Redox Precision Medicine

The development of multi-omics data (genomics, proteomics, metabolomics, transcriptomics) provides new opportunities for multi-network-based biomarker discovery through precision in redox medicine. However, no matter how exciting it may be, it is challenging to quantitatively assess redox-sensitive genomics, proteomics, and metabolomics. AI-based platforms can discover potential biomarkers and, using the curated networks, find potential therapeutic targets. As an example, if we collected data on oxidized proteins (e.g., carbonylated tau), mitochondrial dysfunction (e.g., CoQ10 reduced), or the potential for genetic polymorphic variants in antioxidant enzymes (e.g., SOD2, GPx1, etc.), a patient-centric therapeutic transgenerational pathway could be suggested [282]. Going forward, clinical trials would use digital twin technology to make an individual’s virtual simulations that would combine the participants biomarker profiles (e.g., proteomic, metabolomic), imaging data (e.g., MRI, PET), and the outcomes we measure or look for—coming from the participant’s responses to the acute or chronic therapeutic interventions. Using the digital twin, researchers could prospectively simulate many therapeutic combinations in silico, optimizing the personalized and precise management of the participant’s condition prior to receiving systemic therapy through clinical examination [341].

The future management of oxidative stress mechanism(s) will be multi-modal, multi-targeted, and precise interventions. Merging adaptive expertise through AI-based personalization in therapeutic management, using new nanomedicine-based interventions, and implementing gene-editing technologies would allow for a cure treatment to offset or reverse the disease trajectory. As biomarker-based clinical trials or adaptive clinical trials take center stage and supplant the standard method to manage disease with absolute inevitability for individuals to personalize redox and oxidative stress management strategies, we are in the early days of personalized redox medicine, and patients’ outcome improvements will be transformed immeasurably [342].

## 8. Conclusions: Redefining the Battle Against Oxidative Stress

We are on the verge of a shift in the approach to neurological diseases, and the science has shifted from slowing symptoms or progression; we are pursuing prevention, reversal, and ultimately, cures. For decades, oxidative stress has served as the “hub” of neurodegeneration, activating mitochondrial dysfunction, promoting the formation of misfolded proteins, recruiting neuro-inflammation, and resulting in synaptic dysfunction and cell-death pathways, like the hypoxic and increased lactate pathways. It has operated as a trigger and as the executioner of accelerated progression (e.g., AD; PD; ALS; MS; stroke). Through innovative advances and research, we are re-writing this story today.

In this paper we aimed to explore the most promising therapeutic avenues, including AI-driven precision medicine; combination therapies; mitochondria-targeting antioxidants; gene editing; and localized nanomedicine-based delivery systems. The future is not only promising; the future is inevitable. We have shown that the tools to tackle oxidative stress already exist; it is only a matter of using these tools in a context that can capitalize on these innovations through simultaneous clinical translation and cross-disciplinary collaboration. The previous clinical trials of NAC, CoQ10, MitoQ, and DMF created the backstory and have specifically demonstrated applicable improvements in oxidative damage, mitochondrial function, and clinical outcomes in real-world conditions. NAC has reversed redox balance in dopaminergic neurons by reducing motor decline in patients with PD while preserving cognitive function in patients at risk for AD. MitoQ has rescued ATP production through its targeting action in mitochondria, and sustained synaptic failure at the powerhouse level, but provided additional neuroprotective effects as well. It is obvious, however, that the limitations of monotherapies are clear. One single agent cannot effectively address the tangled dependency chain of all redox imbalances, compete with protein aggregation, or the more aggressive neuro-inflammatory responses. The realization that it is feasible to create mitochondrial-specific antioxidants has heralded a new era of combination therapies, as it is already becoming clear that when used with Nrf2 activators, they show synergistic actions in preclinical studies, provide two levels of protection through reducing ROS in the mitochondria, and stimulating endogenous antioxidant pathways. For instance, MitoQ with DMF can both protect mitochondrial respiration and activate cytosolic detoxification pathways due to cytosolic signaling effects, and each agent provides redox protection. These combinations are already moving into clinical studies related to patients with Parkinson’s disease, stroke, and ALS, as the motivation to deliver something new has never been stronger.

Interestingly, after antioxidant therapies have developed, we now have a potential future role for nanomedicine and truly targeted specificity. Redox-responsive nanoparticles might have the ability to release their therapeutic payload in a target-specific manner in areas of oxidative stress, which already has the potential to change how we target free radical scavengers, anti-inflammatory agents, and even gene therapies. In fact, nanoparticles that were encapsulating CoQ10, MitoQ, and sulforaphane have indicated improved BBB penetrance with ROS-mediated selective de-release of drugs in localized brain areas, to achieve reductions in oxidative damage response levels that are unprecedented while utilizing minimal systemic toxicity. Being able to deliver compounds that target oxidative stress at the time of the attack (in the mitochondria and synapses) is advancing quickly with technology.

However, we are not just satisfied with controlling ROS; we want to approach remapping the genetic programming of a redox disease. Gene-editing methods utilizing CRISPR-Cas9-based technologies might provide a way to permanently correct genetic variations in the SOD1, GPx1, and PRDX3 genes, and assist in remapping a contorted redox population at the genomic level. In familial ALS, where SOD1 mutations heighten oxidative impairment, CRISPR-directed gene correction has triumphed in a preclinical model with reductions in motor neuron death and extended lifespans for the ALS-affected. Furthermore, epigenetic reprogramming with histone deacetylase inhibitors and DNA demethylating agents has led to the reactivation of silenced antioxidant response genes, such as Nrf2, providing enduring protection from neurodegeneration. But, there is a constant need for biomarker-guided precision interventions to assure widespread clinical success globally, in all populations. Any oxidative stress is heterogeneous in terms of between- and among-patient populations. Therefore, AI-powered multi-omics analysis is required to help identify where patients are on their individual redox spectra since oxidative stress is known to be a product of genetic risk, mitochondrial dysfunction, and real-time oxidative stress measures. AI-enabled platforms will define the best financially driven treatment combinations on a patient-specific basis; future clinical trials will not be using outdated protocols; rather, protocols will shift the treatment regimens during the trial to allow for maximum efficacy of a treatment as the disease progresses. But let us be clear, we will still have obstacles. The road to mitigate oxidative stress still has some bumps. Obtaining consistent blood–brain barrier penetration, extending the sustainment of gene therapies, and dosing combinations will all be areas of active research. However, these obstacles will ultimately be overcome. When we consider the velocity of science and technology, guess what the future of oxidative stress will look like; it is not a possibility but rather a reality. If we are successful, wait until you imagine what we will be able to do! Early detection of oxidative stress based on biomarker panels implements prevention long before neurodegeneration ensues. Patients receiving a nanoparticle-delivered antioxidant cocktail to specific brain regions of disease, with sparing of healthy tissue. Gene-edited neurons with localized permanent restored potassium defenses. This is not fiction, friends, this is the science and the reality we are building. We are perched on top of something groundbreaking, and the battle against oxidative stress is no longer relegated to being a steep-nosed hill; it is a bold scientific mission. With the science of AI-powered precision medicine; nanotechnology; and gene-editing, we will take oxidative stress from being an inevitable enemy to being a manageable target, if not something entirely defeated. The question we have to answer is not if we will win, but rather how fast. This paper has disclosed that there are tremendous successes already mapped out, and the speed of success will be phenomenal.

As we conclude, we want to leave the academic reader contemplating the following: The next wave of breakthroughs is almost here, and we are barely on the edge of a new frontier that will commercially transform our field. The oxidative stress management field is no longer about managing to purely lengthen life, but now about restoring life fully again, boundless and beyond cognitive fatigue. Our journey is just beginning. And when the cures come, and they will, these will be the developments we look back on as the shift in transcendence for science daring to overcome the unconquerable.

Let us finish the job.

## Figures and Tables

**Figure 1 ijms-26-07498-f001:**
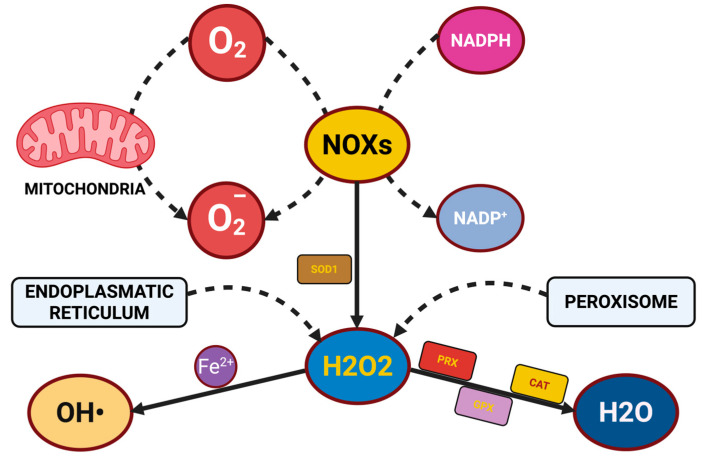
Overview of intracellular reactive oxygen species (ROS) generation and antioxidant detoxification. Superoxide (O_2_^−^) is generated via electron leakage in the mitochondrial electron transport chain (ETC) and by NADPH oxidases (NOXs), notably NOX2 in microglia. SOD1 converts O_2_^−^ into hydrogen peroxide (H_2_O_2_), which is further neutralized by catalase (CAT), glutathione peroxidase (GPX), and peroxiredoxins (PRX) in the cytosol and peroxisomes. In the presence of Fe^2+^, H_2_O_2_ undergoes the Fenton reaction to generate highly reactive hydroxyl radicals (•OH). This figure intends to integrate the major subcellular sources of ROS and highlights the enzymatic antioxidant systems responsible for maintaining redox homeostasis in the brain.

**Figure 2 ijms-26-07498-f002:**
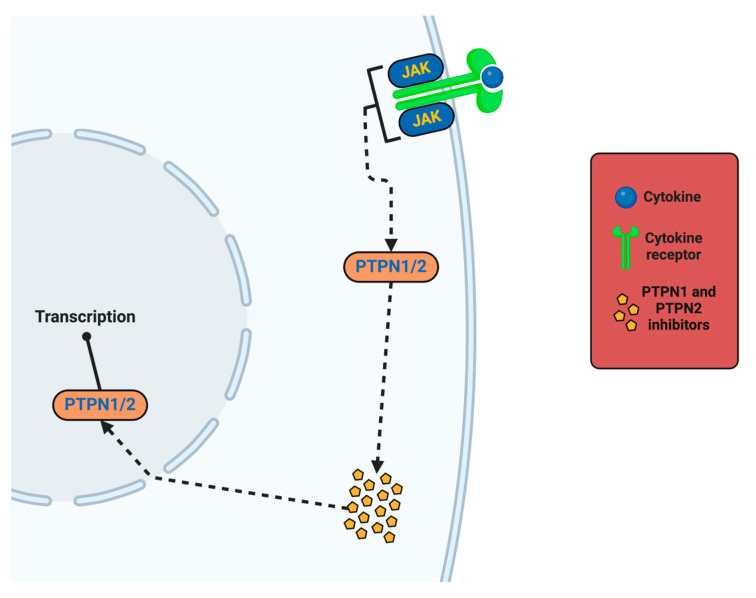
This illustrates the role of PTPN1 and PTPN2 inhibitors in regulating cytokine-induced JAK-STAT signaling pathways. Under normal conditions, cytokine binding to its receptor activates the JAK kinases, which phosphorylate downstream targets and drive the transcription of pro-inflammatory genes. PTPN1 and PTPN2, acting as protein tyrosine phosphatases, negatively regulate this pathway by dephosphorylating key signaling intermediates, effectively dampening the inflammatory response. The inhibition of PTPN1/2 prevents their dephosphorylation activity, thereby sustaining JAK-STAT activation and enhancing the production of inflammatory cytokines. This dysregulated signaling is linked to chronic inflammation and oxidative stress, which contribute to neurodegeneration in diseases like ALS, AD, and PD. However, targeted inhibition of PTPN1/2 offers a novel therapeutic approach to restore redox balance by suppressing cytokine-driven inflammation.

**Figure 3 ijms-26-07498-f003:**
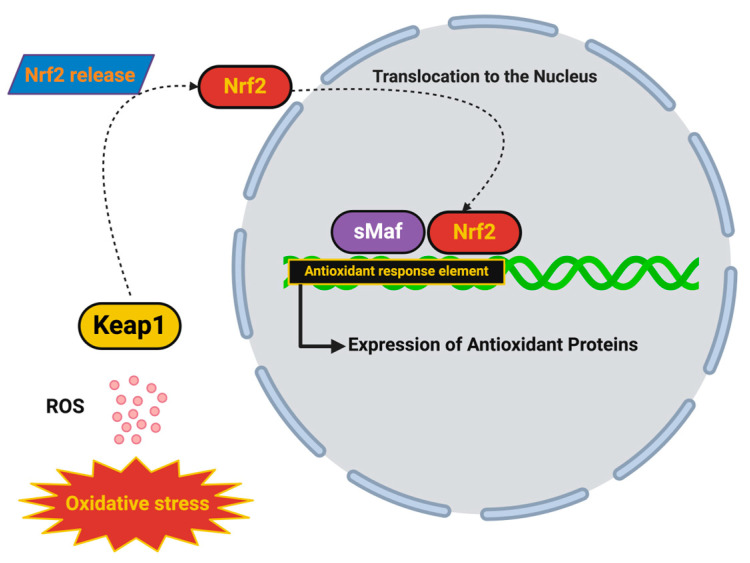
This illustrates the Nrf2-Keap1 pathway activation and Nrf2 nuclear translocation in response to oxidative stress. Under normal conditions, Nrf2 is sequestered in the cytoplasm by its inhibitor, Keap1, which facilitates Nrf2 degradation. However, upon exposure to elevated levels of ROS, oxidative modifications to Keap1 result in the release of Nrf2. Freed from Keap1, Nrf2 translocates to the nucleus, where it forms a complex with small Maf (sMaf) proteins and binds to the antioxidant response element (ARE) within the promoter regions of target genes. This binding initiates the upregulation of antioxidant proteins such as SOD, catalase, GPx, and heme oxygenase-1 (HO-1), which collectively neutralize ROS and restore redox homeostasis.

**Table 1 ijms-26-07498-t001:** A comprehensive summary of key biomarkers of oxidative stress linked to neurological disorders. The biomarkers included represent major classes of oxidative damage—lipid peroxidation (e.g., F2-isoprostanes, 4-HNE), DNA oxidation (e.g., 8-OHdG), protein oxidation and nitration (e.g., nitrotyrosine, protein carbonyls), and indicators of antioxidant capacity (e.g., GSH/GSSG ratio, SOD activity).

Biomarker	Type of Damage	Associated Neurological Disorders	Biological Significance	Detection Method	Reference
F2-isoprostanes	Lipid peroxidation products	AD, PD, Stroke	Are stable and reliable markers of free radical-induced lipid peroxidation; correlate with cognitive decline and neuroinflammation	LC-MS/MS (liquid chromatography–tandem mass spectrometry), GC-MS (gas chromatography-mass spectrometry)	[75]
4-Hydroxynonenal (4-HNE)	Lipid peroxidation byproduct	AD, PD, ALS, Stroke	Is a highly reactive aldehyde that forms covalent adducts with proteins and DNA; promotes protein misfolding and neurotoxicity	ELISA, HPLC, mass spectrometry	[76]
Malondialdehyde (MDA)	Lipid peroxidation marker	AD, PD, ALS, Schizophrenia	Indicates oxidative damage to cell membranes; correlates with mitochondrial dysfunction and neurodegeneration	TBARS (thiobarbituric acid reactive substance) assay, HPLC, spectrophotometry	[77,78]
8-Hydroxy-2′-deoxyguanosine (8-OHdG)	DNA oxidation marker	AD, PD, ALS, Stroke, MS	Indicates oxidative damage to nuclear and mitochondrial DNA; correlates with disease severity and progression when elevated levels are present	ELISA, HPLC, electrochemical biosensors	[79,80,81]
Nitrotyrosine	Protein nitration marker	PD, AD, ALS, Ischemic Stroke	Is a marker of peroxynitrite-induced oxidative stress; is associated with nitrated α-synuclein in PD and neuroinflammation	ELISA, Western blot, immunohistochemistry	[82,83]
Protein Carbonyls	Oxidatively modified proteins	AD, PD, ALS, MS	Indicates irreversible oxidation of proteins, resulting in enzyme inactivation, aggregation, and neurotoxicity	DNPH (2,4-Dinitrophenylhydrazine) derivatization assay, ELISA	[84,85]
Glutathione (GSH/GSSG ratio)	Antioxidant capacity marker	AD, PD, Schizophrenia	Reflects intracellular redox balance; indicates oxidative stress and impaired detoxification when the ratio is low	HPLC, Fluorescence-based assays	[86,87]
Coenzyme Q10 (oxidized form)	Mitochondrial dysfunction marker	PD, ALS, AD	Reduced CoQ10 levels correlate with impaired electron transport chain function and ATP production	HPLC, LC-MS/MS	[88,89]
SOD, Catalase, GPx Activity	Antioxidant enzyme activity	AD, PD, MS, ALS	Decreased activity reflects impaired detoxification of ROS and contributes to sustained oxidative stress	Spectrophotometric enzyme assays, ELISA	[90,91]

**Table 2 ijms-26-07498-t002:** These trials underscore the critical role of redox-targeted therapies in improving clinical outcomes. As emerging evidence suggests, combination therapies and personalized, biomarker-driven interventions hold the greatest potential for future breakthroughs.

Therapeutic Strategy	Agent/Intervention	Clinical Trial Name/Phase	Targeted Mechanism	Disease/Condition	Key Outcomes	Advantages	Limitations	Reference
Mitochondria-Targeted Antioxidant	MitoQ (Mitochondria-targeted CoQ10 derivative)	Phase II PD Trial	Scavenges mitochondrial ROS, protects ETC complexes, and improves ATP production	PD	Reduced oxidative biomarkers (F2-isoprostanes, 4-HNE), improved motor function, and preserved dopaminergic neurons	High mitochondrial selectivity, direct ROS scavenging, and clinical feasibility	Variable efficacy across individuals and limited BBB permeability	NCT00329056 [309]
Nrf2 Activator	Dimethyl fumarate (DMF)	DEFINE and CONFIRM Trials (MS); Phase II ALS	Activates Nrf2, induces antioxidant enzyme production (SOD, catalase, GPx), and reduces lipid peroxidation	Multiple Sclerosis (MS), ALS (repurposed)	Reduced relapse rates and slowed disability progression (MS); ongoing ALS trial testing reduced oxidative damage	Oral availability and upregulation of broad antioxidant response	Possible GI side effects and immunomodulatory concerns	NCT02959658 [310]
Glutathione Precursor	NAC	PD; Cognitive Decline Study (MCI)	Replenishes intracellular GSH, neutralizes ROS, and prevents protein and lipid peroxidation	PD, Mild Cognitive Impairment (MCI)	Improved motor outcomes in PD, preserved cognitive function in MCI, and reduced oxidative stress biomarkers	Clinically approved, good safety profile, and restores GSH levels	Limited CNS bioavailability and requires long-term administration	NCT01470027 [311]
Mitochondrial Membrane Stabilizer	SS-31 (Elamipretide)	Phase II ALS Trial	Binds to cardiolipin, stabilizes mitochondrial membrane potential, and prevents mPTP opening	ALS, Stroke	Improved mitochondrial bioenergetics, reduced muscle weakness, and delayed ALS progression	Targets mitochondrial structure and function directly, and preserves neuromuscular performance	Still in trials, unknown long-term safety, and cost-intensive	NCT05168774 [293]
Nanoparticle-Based Delivery	Nanoparticle-encapsulated CoQ10	Preclinical (PD/AD models)	BBB penetration, targeted release at ROS hotspots, and sustained antioxidant action	Parkinson’s Disease, Alzheimer’s Disease	Enhanced antioxidant efficacy, reduced oxidative stress, and greater preservation of synaptic function	Overcomes BBB limitations, and localized action minimizes systemic toxicity	Still preclinical and requires advanced formulation techniques	NCT01408680 [312]
Epigenetic Therapy	Valproic acid (HDAC inhibitor)	Preclinical AD/PD models	Reactivates silenced antioxidant genes and restores Nrf2 expression	AD, PD	Reduced amyloid burden, improved cognitive outcomes, and reactivated endogenous antioxidant defenses	Modifies gene expression and long-lasting effect on redox balance	Epigenetic off-target effects and dose-dependent toxicity	NCT04698525 [313]
Gene Therapy	CRISPR-Cas9 for SOD1 mutations	Preclinical ALS models	Corrects inherited mutations driving oxidative damage	Familial ALS	Reduced ROS, delayed motor neuron death, and extended survival	Permanent correction of genetic defects and disease-modifying potential	Ethical and safety concerns and high technical complexity	NCT01083667 [314]

## Data Availability

The data presented in this study are available upon request from the corresponding author.

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
