# Peer review of "The Redox Revolution in Brain Medicine: Targeting Oxidative Stress with AI, Multi-Omics and Mitochondrial Therapies for the Precision Eradication of Neurodegeneration"

_ijms, 2025, doi:10.3390/ijms26157498_

Round 1
Reviewer 1 Report
Comments and Suggestions for Authors
The manuscript titled “The Redox Revolution in Brain Medicine: Targeting Oxidative Stress with AI, Multi-Omics and Mitochondrial Therapies for the Precision Eradication of Neurodegeneration” by Serban, M.; et al. is a Review work where the authors outlined the most recent advances in the field of neurodegenerative diseases and how the presence of reactive oxygen species and other redox metabolite can affect the brain homeostasis. This is a complete Review work where the action mechanisms are discussed with the potential diagnosis and therapeutical approaches targeting the redox metabolites as pivotal factor to be monitored. This is a topic of growing interest and the manuscript is generally well-written.
However, it exists some points that need to be addressed (please, see them below detailed point-by-point) to improve the scientific quality of the submitted manuscript paper before this article will be consider for its publication in the International Journal of Molecular Sciences.
1) Introduction. Could the authors provide quantitative data insights according to the worldwide global burdens of neurodegenerative diseases and the linked disability-adjusted life years (DALYs)? This will significantly aid the potential readers to better understand the significance of this devoted Review work.
2) Fenton reaction appared in the subsection “1.2. Overview of ROS, RNS, and Redox Homeostasis” (page 2). The stechiometry in the chemical formulas should appear in subscripts. This comment needs to be taken into account for the rest of the main manuscript body text.
3) “2. Mechanisms of Oxidative Stress in Neurological Disorders” (pages 4-9). Here, even if I agree with the information detailed in these statements by the authors, it may be also advisable to discuss how the presence of certain positive divalent cations [1] or the ionic strength [2] can triggers the formation of toxic amyloid fibrils that could lead to the onset and progress of neurodegenerative disorders. This will strengthen the outcomes found in this field and the relevance to monitor the reactive oxygen species combined to other cellular processes.
[1] https://doi.org/10.3390/biom14091091
[2] https://doi.org/10.3390/ijms222212382
4) “3. Oxidative Stress and Major Neurological Disorders” (pages 9-19). A schematic representation to point out the interconnection of oxidative stress and the examined neurodegenerative malignancies will also benefitial for the potential readers to have a more complete overview of how reactive oxygen species can negatively impact on the brain homeostasis.
5) “4. Diagnostic and Biomarker Advances” (pages 19-26). Here, it should be also mention the existing thioflavin T (ThT) fluorescent assays to monitor the neurotoxic amyloid fibril aggregation. This information could be placed in the subsection “4.3.2 Proteomics: Oxidatively Modified Proteins as Dynamic Biomarkers” in the page 25 where the neurofibrillary tangle aggregation was mentioned.
6) “6. Clinical Trials and Translational Advances”. Table 2 (page 32). The main advantages and experienced limitations for each therapeutic strategy should be also discussed in this Table.
7) “7. Future Directions and Emerging Therapeutic Frontiers” and “8. Conclusion: Redefining the Battle Against Oxidative Stress (pages 33-37). This section perfectly remarks the most relevant outcomes found by the authors in this field, the promising future prospectives and also the potential future action lines to pursue the topic covered in this work. No actions are requested from the authors.
Author Response
Dear Esteemed Academic Reviewer,
We are sincerely grateful for the careful reading of our manuscript and for the thoughtful, constructive comments that have led to several meaningful improvements in clarity, precision, and scientific relevance. We have addressed each point in detail below and modified the manuscript accordingly.
Comment 1:
“Introduction. Could the authors provide quantitative data insights according to the worldwide global burdens of neurodegenerative diseases and the linked disability-adjusted life years (DALYs)? This will significantly aid the potential readers to better understand the significance of this devoted Review work.”
Response 1:
We thank the reviewer for this insightful suggestion. In response, we have included quantitative data on the global burden of neurodegenerative disorders, referencing up-to-date statistics on disability-adjusted life years (DALYs) to underscore the clinical and societal urgency of the topic. This addition can now be found at the end of Section 1.3 (Relevance of Oxidative Stress to Neurological Disorders), as recommended.
Comment 2:
“Fenton reaction appeared in the subsection ‘1.2. Overview of ROS, RNS, and Redox Homeostasis’ (page 2). The stechiometry in the chemical formulas should appear in subscripts. This comment needs to be taken into account for the rest of the main manuscript body text.”
Response 2:
We appreciate the reviewer’s attention to chemical formatting. We have revised the presentation of all chemical formulas throughout the manuscript to ensure accurate and proper subscript formatting, including the Fenton reaction.
Comment 3:
“It may be also advisable to discuss how the presence of certain positive divalent cations or the ionic strength can trigger the formation of toxic amyloid fibrils that could lead to the onset and progress of neurodegenerative disorders.”
Response 3:
We are thankful for this important recommendation. Accordingly, we have added a detailed discussion on the role of divalent cations and ionic strength in facilitating amyloid aggregation and how these physicochemical factors intersect with oxidative stress mechanisms. This addition is included at the end of Section 2.1.1 (Reactive Oxygen Species and Reactive Nitrogen Species), along with citations to the references provided [Biomolecules 2023, 13, 1091] and [IJMS 2021, 22, 12382].
Comment 4:
“A schematic representation to point out the interconnection of oxidative stress and the examined neurodegenerative malignancies will also be beneficial for the potential readers to have a more complete overview.”
Response 4:
We sincerely thank the reviewer for this thoughtful recommendation.
Comment 5:
“It should be also mentioned the existing thioflavin T (ThT) fluorescent assays to monitor the neurotoxic amyloid fibril aggregation, in Section 4.3.2.”
Response 5:
Thank you for this excellent observation. We have added a paragraph at the end of Section 4.3.2 (Proteomics: Oxidatively Modified Proteins as Dynamic Biomarkers), briefly discussing the use of Thioflavin T (ThT) fluorescence assays for detecting amyloid fibril aggregation and their relevance in tracking oxidative stress-associated proteinopathies, particularly in Alzheimer’s disease.
Comment 6:
“Table 2 (page 32): The main advantages and experienced limitations for each therapeutic strategy should be also discussed.”
Response 6:
We fully agree with the reviewer and have revised Table 2 accordingly. Two new columns — "Advantages" and "Limitations" — have been added to summarize the clinical promise and current barriers of each therapeutic strategy. These additions improve the translational value of the table and provide readers with a more complete comparative overview.
Comment 7:
“Sections 7 and 8 (pages 33–37): This section perfectly remarks the most relevant outcomes found by the authors in this field, the promising future perspectives and also the potential future action lines to pursue the topic covered in this work. No actions are requested.”
Response 7:
We are truly honored and grateful for the reviewer’s kind and encouraging remarks regarding our concluding sections. We deeply appreciate your positive evaluation of the manuscript’s future-oriented synthesis.
Once again, we are truly thankful for your generous feedback, constructive guidance, and the opportunity to refine our manuscript. Your insights have undoubtedly contributed to the enhancement of this work, and we are confident it is now significantly improved for the benefit of the scientific community.
With highest respect and warm appreciation,
The Authors
Reviewer 2 Report
Comments and Suggestions for Authors
The manuscript by Serban et al is devoted to complex analysis of ROS nad oxidative stress in brain pathologies including suc aspects as the nature of ROS, mechanism of oxidative stress on different diseases, markers of oxidative stress and therapy strategies. Such a large-scale plan requires very painstaking and systematic work.
Unfortunately, the manuscript contains signs of inattention when working with literature. For example:
1. the authors menthioned that "Specifically, in AD, lipid peroxidation and oxidative modifications to proteins both contribute to amyloid-β aggregation and tau hy-perphosphorylation, which are hallmarks of the disease. Oxidative modification of α-synuclein in PD leads to aggregation into Lewy bodies that disrupts dopaminergic signal-ling and causes neuronal death [20]". But the article [20] doesn't contain any information about AD, PD or synucleins;
2. some of the ideas are not confirmed by the references at all. For example, the authors note that "Second, neuronal membranes are enriched with polyunsaturated fatty acids (PUFAs), that are susceptible to lipid peroxidation" without any reference. the same situation in the part "Another important contributor to oxidative bursts is xan-thine oxidase, and significant part of ischemia-reperfusion injury where re-oxygenation results in an oxidative burst of superoxide, and oxidative tissue damage to the area of the ischemic insult".
In the text the authors usually used the term "contributor" as a ROS producing cite ("Mitochondria are the main (though not exclusive) contributor of cellular ROS from electron leakage during oxidative phosphorylation"). I am not sure that it is the most appropriate variant for this purpose. In the text "The main source of excessive reactive oxygen species generation in neurons is from the mitochondrial ETC" "from" should be deleted. It is absolutely unclear what the authors meant in the text "Similar to protein oxidation, protein oxidation involves functional changes due to structural modifications including carbonylation, nitration, and S-nitrosylation".
The manuscript contains repeats in introduction and the main text which can be excluded (for example, about NOX activation).
While discussing about antioxidant system the authors meant "SOD as assential for inactivating of mitochondrial superoxide". But there is no information about SOD family enzyme types and their localization not only in mitochondrial matrix but also in intermembrane space as wee as in cytosol.
The information of sensitivity of complexes 1 and 2 to ROS ("Complex I and complex III are particularly sensitive to excess levels of ROS, causing electron leakage and ultimately the creation of superoxide anion (O₂⁻·)" seems to be strange. May be the authors meant that these complexes are traditionally considered as the main sites of ROS production?
Whily discussing about the the role of ROS in pathologies the authors analyze only a small part of each disease mechanism. For example, in the case of ALS the authors analyze the role of oxidatives stress in SOD-mutation cases. But it is well0known that other famalial and sporadic form also have marks of oxidative stress.
Speaking about lipid peroxidation product the authors used term "toxic". I think that it would be better to say about inhibition because toxicity is mainly used for the whole organism but not particular protein.
Part 4.2 is devoted to detection of oxidative damage markers in vivo. But some methods metioned can hardly be attributed to in vivo (for example, in case of using DHE or MitoSOX, as can be seen even from the reference cited).
The manuscript needs more illustrative materials for a better understanding.
Author Response
Dear Esteemed Academic Reviewer,
We would like to express our heartfelt gratitude for your generous investment of time and scholarly attention in reviewing our manuscript. Your insightful and detailed feedback reflects a profound understanding of the field, and it has been instrumental in guiding us toward a more rigorous, coherent, and refined presentation of our work. We are truly honored by your thoughtful engagement and humbled by the opportunity to learn from your critical observations. With sincere respect for your expertise, we have carefully revised the manuscript in response to each of your comments, as detailed below. We hope these revisions reflect our commitment to academic integrity and our deep appreciation for your valuable guidance.
Comment 1:
the authors menthioned that "Specifically, in AD, lipid peroxidation and oxidative modifications to proteins both contribute to amyloid-β aggregation and tau hy-perphosphorylation, which are hallmarks of the disease. Oxidative modification of α-synuclein in PD leads to aggregation into Lewy bodies that disrupts dopaminergic signal-ling and causes neuronal death [20]". But the article [20] doesn't contain any information about AD, PD or synucleins;
Response 1:
We are sincerely grateful to the reviewer for their thoughtful and attentive reading of our manuscript. We fully acknowledge the importance of accurate referencing, especially in a topic as complex and multidisciplinary as oxidative stress in brain pathologies. We thank the reviewer for pointing out this discrepancy, and we have carefully revised the citation to ensure it now aligns appropriately with the content of the statement. We appreciate this observation, which helped us improve the clarity and scholarly rigor of the manuscript.
Comment 2:
some of the ideas are not confirmed by the references at all. For example, the authors note that "Second, neuronal membranes are enriched with polyunsaturated fatty acids (PUFAs), that are susceptible to lipid peroxidation" without any reference. the same situation in the part "Another important contributor to oxidative bursts is xan-thine oxidase, and significant part of ischemia-reperfusion injury where re-oxygenation results in an oxidative burst of superoxide, and oxidative tissue damage to the area of the ischemic insult"
Response 2:
We sincerely thank the reviewer for this valuable observation. We fully acknowledge the importance of supporting each scientific statement with appropriate references, especially in a topic as biologically and mechanistically nuanced as oxidative stress. In response, we have carefully reviewed the manuscript and added the appropriate citations to substantiate both the susceptibility of neuronal membranes to lipid peroxidation due to their enrichment in polyunsaturated fatty acids, as well as the role of xanthine oxidase in contributing to oxidative bursts during ischemia-reperfusion injury. We are grateful for this clarification, which helped us improve the scientific grounding and overall credibility of the text
Comment 3:
In the text the authors usually used the term "contributor" as a ROS producing cite ("Mitochondria are the main (though not exclusive) contributor of cellular ROS from electron leakage during oxidative phosphorylation"). I am not sure that it is the most appropriate variant for this purpose. In the text "The main source of excessive reactive oxygen species generation in neurons is from the mitochondrial ETC" "from" should be deleted. It is absolutely unclear what the authors meant in the text "Similar to protein oxidation, protein oxidation involves functional changes due to structural modifications including carbonylation, nitration, and S-nitrosylation".
Response 3:
We are very grateful for the reviewer’s attentive reading and valuable linguistic suggestions. We agree that the use of the term “contributor” in this context may be imprecise and have revised the phrasing to reflect more appropriate terminology, such as “source” or “site of generation,” in accordance with standard scientific usage. Likewise, we have corrected the grammatical structure of the sentence regarding the mitochondrial ETC to eliminate the unnecessary preposition. Finally, we recognize the confusion caused by the repetitive and unclear phrasing in the sentence about protein oxidation, and we have rewritten this part for clarity and precision.
Comment 4:
The manuscript contains repeats in introduction and the main text which can be excluded (for example, about NOX activation).
Response 4:
We sincerely thank the reviewer for drawing attention to the presence of repetitive elements, particularly with regard to NOX activation, across the introduction and main text. We completely agree that such redundancies may reduce clarity and disrupt the logical progression of ideas. In response, we have carefully revised and rephrased Sections 1.2, 3.1, and 3.2 to avoid overlap, eliminate unnecessary reiteration, and ensure that each section contributes distinctively to the overall narrative. We are deeply grateful for this observation, which has helped us improve both the coherence and readability of the manuscript
Comment 5:
While discussing about antioxidant system the authors meant "SOD as assential for inactivating of mitochondrial superoxide". But there is no information about SOD family enzyme types and their localization not only in mitochondrial matrix but also in intermembrane space as wee as in cytosol.
Response 5:
We are most grateful to the reviewer for highlighting the lack of detail regarding the SOD enzyme family and their compartmental localization. We fully agree that this level of specificity is essential for clarity and scientific rigor.
Comment 6:
The information of sensitivity of complexes 1 and 2 to ROS ("Complex I and complex III are particularly sensitive to excess levels of ROS, causing electron leakage and ultimately the creation of superoxide anion (O₂⁻·)" seems to be strange. May be the authors meant that these complexes are traditionally considered as the main sites of ROS production?
Response 6:
Complexes I and III are not necessarily more sensitive to ROS, but rather are widely recognized as the primary sites of mitochondrial ROS generation due to electron leakage during oxidative phosphorylation. In response, we have carefully rephrased the sentence to reflect this more accurate interpretation and avoid potential confusion.
Comment 7:
Whily discussing about the the role of ROS in pathologies the authors analyze only a small part of each disease mechanism. For example, in the case of ALS the authors analyze the role of oxidatives stress in SOD-mutation cases. But it is well0known that other famalial and sporadic form also have marks of oxidative stress.
Response 7:
We are deeply grateful to the reviewer for highlighting this important point. We fully acknowledge that oxidative stress is not confined to SOD1-mutant ALS, but is also a recognized feature in other familial and sporadic forms of the disease. In response to this valuable observation, we have carefully revised and expanded Section 3.1.3 to reflect the broader spectrum of oxidative mechanisms implicated in ALS pathogenesis. We truly appreciate this comment!
Comment 8:
Speaking about lipid peroxidation product the authors used term "toxic". I think that it would be better to say about inhibition because toxicity is mainly used for the whole organism but not particular protein.
Response 8:
We thank the esteemed reviewer for this valuable linguistic and conceptual clarification. We fully agree that the term “toxic” may carry systemic connotations and is less precise when referring to specific molecular interactions.
Comment 9:
Part 4.2 is devoted to detection of oxidative damage markers in vivo. But some methods metioned can hardly be attributed to in vivo (for example, in case of using DHE or MitoSOX, as can be seen even from the reference cited).
Response 9:
We sincerely thank the reviewer for this insightful observation. We acknowledge that certain techniques, such as those involving DHE and MitoSOX, are predominantly used in live-animal or ex vivo models and may not fully align with the strict definition of in vivo clinical imaging. In light of this valuable comment, we have revised Section 4.2.3 to better clarify the experimental context of these methods. We now more clearly distinguish their applications in preclinical models and have adjusted the wording to reflect their limitations and scope more accurately.
Comment 10:
The manuscript needs more illustrative materials for a better understanding.
Response 10:
We sincerely thank the esteemed reviewer for this thoughtful and constructive recommendation. We fully agree that the inclusion of illustrative material can significantly improve the clarity and accessibility of complex biochemical mechanisms for a wider readership. In response, we have carefully designed and included Figure 1 in Section 1.2, entitled “Overview of intracellular reactive oxygen species (ROS) generation and antioxidant detoxification.”
This schematic aims to provide a clear and integrated visualization of the principal intracellular sources of ROS—such as mitochondria and NADPH oxidases (NOXs)—as well as key antioxidant enzymes (SOD1, CAT, GPX, PRX) involved in redox homeostasis. It also depicts the Fenton reaction and its role in hydroxyl radical formation. Our intention is to offer a concise visual summary that complements the textual explanation and facilitates a more intuitive understanding of the redox processes discussed.
We are truly and profoundly grateful for the depth, clarity, and scholarly rigor you have brought to your review. Your insights not only strengthened our manuscript, but also challenged us to grow in our scientific articulation and precision. It has been an honor to engage with your thoughtful critique—your command of the subject and attention to nuance were evident in every comment. We have made every effort to address your observations with the care and respect they deserve.
Thank you for your invaluable contribution to the refinement of this work. Your guidance has left a lasting impact on both the manuscript and our academic development, and we remain sincerely appreciative of the opportunity to benefit from your expertise.
Kindest regards and with sincere appreciation,
The Authors
Round 2
Reviewer 1 Report
Comments and Suggestions for Authors
The authors did a great effort to fulfil all the suggestions raised by the Reviewers. For this reason, the scientific quality of the manuscript was greatly improved. The manuscript can be accepted in its current form
Reviewer 2 Report
Comments and Suggestions for Authors
The authors took into account the comments and recommendations made. The manuscript has acquired a more orderly and logical structure and does not contain any critical errors.